# REBAR: Retrieval-Based Reconstruction for Time-series Contrastive Learning

**Maxwell A. Xu[1], Alexander Moreno[1], Hui Wei[3], Benjamin M. Marlin[3], James M. Rehg[2]**
[1] Georgia Tech, [2] UIUC, [3] UMass Amherst
`maxxu@gatech.edu, jrehg@illinois.edu`

## ABSTRACT

The success of self-supervised contrastive learning hinges on identifying positive data pairs, such that when they are pushed together in embedding space, the space encodes useful information for subsequent downstream tasks. Constructing positive pairs is non-trivial as the pairing must be similar enough to reflect a shared semantic meaning, but different enough to capture within-class variation. Classical approaches in vision use augmentations to exploit well-established invariances to construct positive pairs, but invariances in the time-series domain are much less obvious. In our work, we propose a novel method of using a learned measure for identifying positive pairs. Our Retrieval-Based Reconstruction (REBAR) measure measures the similarity between two sequences as the reconstruction error that results from reconstructing one sequence with retrieved information from the other. Then, if the two sequences have high REBAR similarity, we label them as a positive pair. Through validation experiments, we show that the REBAR error is a predictor of mutual class membership. Once integrated into a contrastive learning framework, our REBAR method learns an embedding that achieves state-of-the-art performance on downstream tasks across various modalities.

## 1 INTRODUCTION

Self-supervised learning uses the underlying structure within a dataset to learn rich and generalizable representations without labels, enabling fine-tuning on various downstream tasks. This reduces the need for large labeled datasets, which is attractive for many machine learning tasks, and particularly useful in the analysis of time series data for health applications. Due to advances in sensor technology, it is increasingly feasible to capture a large volume of health-related time-series data (Nasiri & Khosravani, 2020), but the cost of labeling this data remains high. For example, in mobile health applications, acquiring labels requires burdensome real-time annotation (Rehg et al., 2017) by participants. Additionally, in medical applications such as ECG analysis, annotation is costly as it requires specialized medical expertise.

Contrastive learning is a powerful self-supervised approach to learning semantically-meaningful representations, which is based on constructing and embedding positive and negative pairs of unlabeled samples. In order to obtain useful representations, pairs should capture important structural properties of the data. In the vision applications that have driven this approach, augmentations are used to construct positive pairs by exploiting invariances of the imaging process (e.g. transformations such as flipping and rotating that change the data vector without changing its meaning). Unfortunately, general time-series do not possess a large and rich set of such invariances. Shifting, which addresses translation invariance, is widely-used, but other augmentations such as shuffling or scaling can destroy the signal semantics. For example, shuffling an ECG waveform destroys the temporal structure of the QRS complex, and scaling it can change the clinical diagnosis (Nault et al., 2009). Moreover, there is no consistent consensus of augmentations in the literature; methods such as TF-C (Zhang et al., 2022) incorporate jittering and scaling, while TS2Vec (Yue et al., 2022) finds that these augmentations impair downstream performance.

In this work, we introduce a novel approach for identifying positive pairs for time-series contrastive learning. Our key idea is that instead of generating positive pairs via augmentation, we use a learned similarity measure to identify positive pairs that naturally occur in extended time-series recordings.

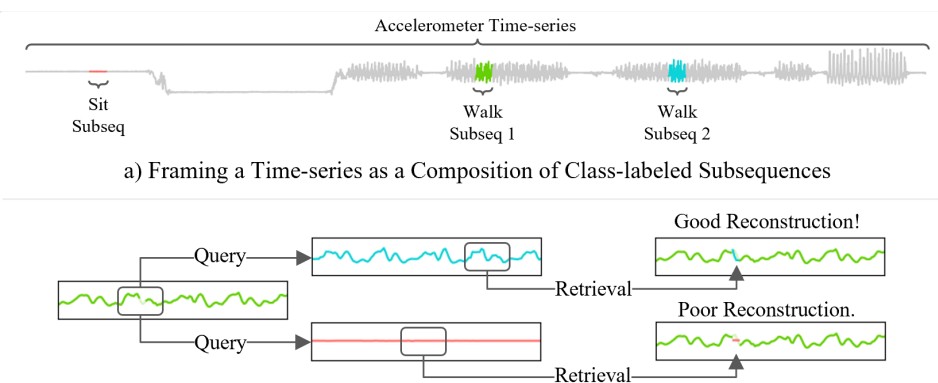

a) Framing a Time-series as a Composition of Class-labeled Subsequences

b) Query other subsequences for information to retrieve to be used to reconstruct the original

Figure 1: This figure demonstrates the intuition of our Retrieval-Based Reconstruction (REBAR) approach. If we can successfully retrieve information from another subsequence to aid in reconstruction, then the two subsequences should form a positive pair in a contrastive learning framework. We first use the context-window, designated by the grey box, of Walk Subseq 1 to query for information in Walk Subseq 2 or in Sit Subseq. Upper b) shows that the context window in Walk Subseq 2 provides a good match with a similar double peak motif, leading to a good reconstruction. Lower b) shows that Sit Subseq has no matching motif, leading to a poor reconstruction.

In our framework, we conceptualize a time-series as a composition of a sequence of subsequences, each of which has a class label. This framing describes many real-world physiological signals. For example, a daily record of an accelerometry signal from a wrist-worn smartwatch contains many repeated subsequences corresponding to frequent activities like walking or sitting. Each subsequence from the same activity class will in turn be comprised of brief temporal patterns or "motifs" such as a "swing up" hand motion during walking. Likewise, the deflections in an ECG signal due to the depolarization of the heart define motifs within the QRS complex (Bouaziz et al., 2014). If two subsequences contain similar motifs, then they are likely to share the same class label and are therefore a good candidate to form a positive pair.

We operationalize the idea of matching motifs with our Retrieval-Based Reconstruction (REBAR)[1] approach. In order to avoid explicitly modeling and detecting motifs, we adopt a reconstruction-based approach in which masked samples in one subsequence are reconstructed directly from values retrieved from a second, candidate subsequence, as illustrated in Fig. 1. A context window is taken around each masked sample, and the REBAR cross-attention model learns to compare each context window to the windows in the candidate subsequence to be retrieved for reconstruction. When two subsequences have many motifs in common, high quality matches can be obtained that minimize reconstruction error. Therefore, the REBAR reconstruction error is a learned measure that captures motif similarity, and pairs with a lower error can then form positive examples in contrastive learning. Such pairs are likely to share semantic meaning, so that the resulting learned embedding space is class-discriminative. We are able to demonstrate this by showing that REBAR achieves state-of-the-art performance on a diverse set of time-series. The full REBAR approach can be seen in Fig. 2, and our public code repository can be found here: `https://github.com/maxxu05/rebar`.

Our main contributions in this work are:

1. This is the first work to use a similarity measure to select positive and negative pairs in time-series contrastive learning. We do so with our REBAR measure, which captures motif-similarity between subsequences using a convolutional cross-attention architecture.
2. We demonstrate that our learned measure predicts mutual class membership in a nearest neighbor sense, which validates that our positive pairs are implicitly capturing the subtle invariances within time series signals, as required for contrastive learning.
3. Our REBAR contrastive learning approach achieves SOTA performance against a representative set of contrastive learning methods that encompass the different ways in which positive and negatives pairs can be generated. Note that our contrastive training method also beats a fully-supervised training approach.

---

[1]Note that we will interchangeably use "REBAR" to refer to the REBAR contrastive learning approach, the REBAR cross-attention, or the REBAR measure. The specific meaning will be evident from the context.

## 2 RELATED WORK

**Augmentation-based Contrastive Learning:** Augmentation-based methods are the most studied type of contrastive learning method in time-series research, due to the success of augmentation-based strategies in computer vision (He et al., 2020; Chen et al., 2020; Chen & He, 2021; Caron et al., 2021). However, it is unclear which augmentation strategies are most effective for time-series, and the findings across different works are inconsistent. TS2Vec (Yue et al., 2022) uses cropping and masking to create positive examples, and their ablation study found that jittering, scaling, and shuffling augmentations led to performance drops. Conversely, TF-C (Zhang et al., 2022) included jittering and scaling, along with cropping, time-shifting, and frequency augmentations. TS-TCC (Eldele et al., 2023) augments the time-series with either jittering+scaling or jittering+shuffling. This is in spite of how shuffling breaks temporal dependencies, and scaling changes the semantic meaning of a bounded signal. Other augmentation works (Woo et al., 2022; Yang & Hong, 2022; Yang et al., 2022b; Ozyurt et al., 2022; Lee et al., 2022) also use some combination of scaling, shifting, jittering, or masking. Empirical performance was used to justify the augmentation choice, but differences in datasets, architectures, and training regimes make it difficult to draw a clear conclusion on what the best set of augmentations are. Our REBAR method instead uses a sampling-based approach to identify positive instances from a set of real subsequences, rather than generating a positive instance from a inconsistent set of augmentations.

**Sampling-based Contrastive Learning:** After sampling an anchor subsequence, TLoss (Franceschi et al., 2019) creates the positive subsequence as a crop of the anchor and the negative as a crop from a different time-series. CLOCS (Kiyasseh et al., 2021) samples pairs of temporally-adjacent subsequences and pairs of subsequences across channels from the same time-series as positives. TNC (Tonekaboni et al., 2021) randomly samples a positive example from the anchor subsequence's neighborhood region and an unlabeled example from outside. The neighborhood is found via a stationarity test, resulting in TNC's run-time being 250x slower than TS2Vec (Yue et al., 2022), and it utilizes a hyperparameter to estimate the probability that the unlabeled example is a true negative. Our REBAR approach is sampling-based, but unlike previous work, our positive examples are not selected based on temporal proximity to the anchor. Instead, positive examples are selected on the basis of their similarity to the anchor, measured by retrieval-based reconstruction.

**Other Self-Supervised Learning Methods:** CPC is a contrastive learning method learns to contrast future points against incorrect ones (Oord et al., 2018). There have also been contrastive learning methods designed for specific sensor modalities with expert knowledge, such as for EEG data (Zhang et al., 2021). Another method is the Masked Autoencoder, which involves masked reconstruction but is fundamentally different from REBAR. See Appendix A.1.5 for further discussion.

**Time-Series Motifs:** A motif is a brief temporal shape that repeats itself approximately across the time-series and is potentially class-discriminative. Much work has been done in identifying motifs via works such as matrix profile (Yeh et al., 2016; 2018; Gharghabi et al., 2018), and there are many classical time-series approaches that use template-matching methods to classify motifs (Frank et al., 2012; Okawa, 2019; Niennattrakul et al., 2012). However, instead of decomposing our time-series into specific motifs as the classical literature does, our REBAR method uses cross-attention to retrieve motifs that are useful in the context of reconstruction. Then, we can utilize the reconstruction error to capture motif-similarity in a novel contrastive learning context for identifying positive pairs.

## 3 NOTATION

The dataset is designated by $\mathbf{A} \in \mathbb{R}^{N \times U \times D}$, with $N$ long time-series of $U$ temporal length and $D$ channels. $\mathbf{A}^{(i)} \in \mathbb{R}^{U \times D}$ is the $i$th time-series in the dataset. $\mathbf{X}^{(i)} \in \mathbb{R}^{T \times D}$ is a subsequence of the $\mathbf{A}^{(i)}$ with length $T$, where $\mathbf{X}^{(i)} = \mathbf{A}^{(i)}[t : t + T]$, for some $t \in \mathbb{N}$ where $T \ll U$. $(i)$ will be omitted for brevity when not relevant. $\mathbf{x} \in \mathbb{R}^{D}$ refers to a specific time-point's data found in $\mathbf{X}$.

Throughout the paper, a subscript is used to describe a specific subsequence, $\mathbf{X}_{\text{description}}$. Within cross-attention, $\mathbf{X}_q$ and $\mathbf{X}_k$ designate the subsequences that serve as the query or key, respectively. A bar in $\bar{\mathbf{X}}$ designates that $\mathbf{X}$ has been partially masked out. In contrastive learning, $\mathbf{X}_{\text{anchor}}$ is the anchor, and $\mathbf{X}_{\text{cand}}$ is a candidate. We then identify which of the candidates $\mathbf{X}_{\text{cand}}$, should be labeled as positive, $\mathbf{X}_{\text{pos}}$, or negative, $\mathbf{X}_{\text{neg}}$.

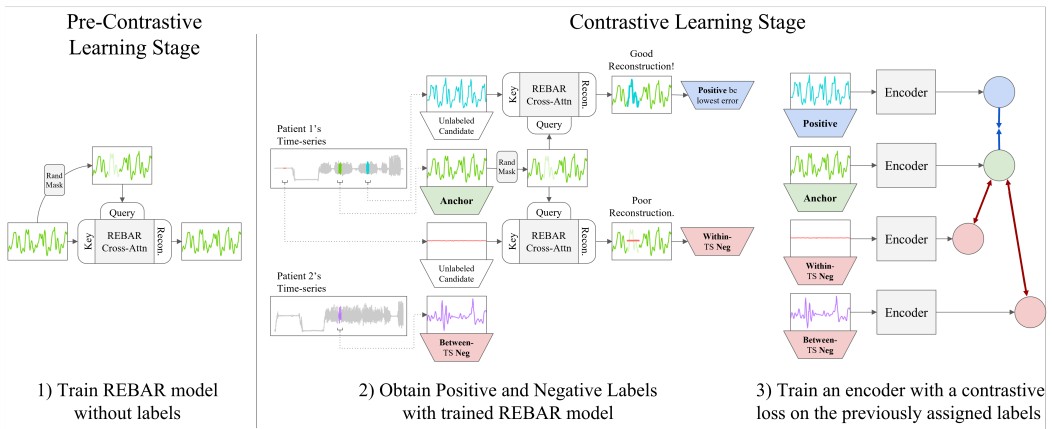

Figure 2: 1) First, our REBAR cross-attention is trained to retrieve information from the key to reconstruct a masked-out query. 2) Next, it is frozen and utilized to identify the positive instance. After sampling subsequences from the time-series, the subsequence that reconstructed the anchor with the lowest REBAR error is labeled as positive, and the others are labeled as within-time-series negatives. These negatives capture how time-series dynamics can change over time. Subsequences from other time-series within a data batch are labeled as between-time-series-negatives, and these negatives capture differences among patients. 3) We use the assigned labels to train an encoder.

## 4 REBAR APPROACH

Self-supervised contrastive learning methods learn an embedding by constructing positive and negative instance pairs and then pushing the positive pairs together and negative pairs apart. In order to construct positive and negative pairs via retrieval, we designate one subsequence as the anchor, and use our REBAR measure to quantify the similarity between the anchor and other instances from the same time-series. The most similar instance forms a positive pair with the anchor, while the other instances, including instances from other time-series, form the negative pairs. Sec. 4.1 describes how we design our REBAR cross-attention module to produce the REBAR measure, and Sec. 4.2 explains how we apply the measure for sequence comparison in a contrastive learning framework. Sec. 4.3 tests the hypothesis that the REBAR measure can capture semantic relationships by demonstrating that REBAR predicts mutual class membership.

$\text{REBAR}(\mathbf{X}_{\text{anchor}}, \mathbf{X}_{\text{cand}})$ cross-attention reconstructs $\bar{\mathbf{X}}_{\text{anchor}}$ by retrieving motifs in $\mathbf{X}_{\text{cand}}$ that match the context window. Then, REBAR error serves as a distance measure[2] between two sequences, shown in Eq. 1. We hypothesize that if $d(\mathbf{X}_{\text{anchor}}, \mathbf{X}_{\text{cand}})$ is small, then it predicts if $\mathbf{X}_{\text{cand}}$ is the same class as $\mathbf{X}_{\text{anchor}}$ (i.e. mutual class membership), allowing us to identify positive pairs.

$$d(\mathbf{X}_{\text{anchor}}, \mathbf{X}_{\text{cand}}) := \|\text{REBAR}(\bar{\mathbf{X}}_{\text{anchor}}, \mathbf{X}_{\text{cand}}) - \mathbf{X}_{\text{anchor}}\|_2^2 \tag{1}$$

### 4.1 DESIGN OF THE REBAR CROSS-ATTENTION

We would like to design our retrieval-based reconstruction error to be class-discriminative, such that pairs with better reconstruction and lower distance are more likely to share classes and thus be semantically related. As such, REBAR identifies the motifs from the candidate, $\mathbf{X}_{\text{cand}}$, that best match to the context window of the anchor, $\bar{\mathbf{X}}_{\text{anchor}}$ and retrieves these motifs to reconstruct the anchor. For example, take the visualizations shown in Fig. 1. The error resulting from reconstructing Walk Subsequence 1 from the retrieved matching motif in Walk Subsequence 2 is lower than reconstructing from the retrieved motif in the Sit Subsequence. The reconstruction performance is dependent on how closely the motifs in the candidate are able to match with the anchor, which allows for the REBAR measure to be class-discriminative.

Cross-attention learns to produce weighted averages of a transformation of the key time-series and is an attractive method for modeling this paradigm. This is because the retrieval function, $p(\mathbf{x}_k|\mathbf{x}_q)$ as shown in Eq. 2, can be interpreted as identifying the $\mathbf{x}_k$ that best matches $\mathbf{x}_q$. Cross-attention is most commonly used with text acting as a query to retrieve relevant regions in an image (Lee et al., 2018a; Miech et al., 2021; Zheng et al., 2022), but it has been used occasionally for supervised

---

[2]We refer to this as a measure because it is not a valid distance metric, violating symmetry+triangle inequality.

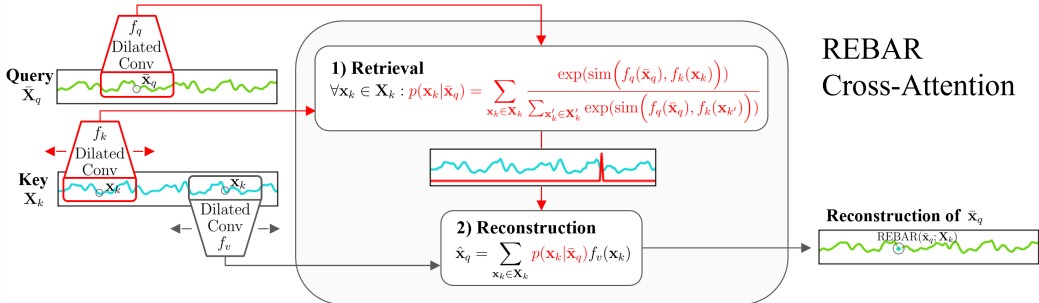

Figure 3: REBAR Cross-attention reconstruction of a single masked-out point $\bar{\mathbf{x}}_q$, REBAR($\bar{\mathbf{x}}_q; \mathbf{X}_k$). Red designates the functions used for the attention weight calculation. In 1), the attention weights identify which region in $\mathbf{X}_k$ should be retrieved for reconstruction by comparing the unmasked context around $\bar{\mathbf{x}}_q$ via the $f_q$ dilated convolution, with the motifs in $\mathbf{X}_k$ via the $f_k$ dilated convolution. In 2), the attention weights are used to retrieve an $f_v$ transformation of the key in a weighted average for reconstruction. Minor model details (e.g. norms) are omitted for brevity.

time-series tasks (Garg & Candan, 2021; Yang et al., 2022a). We describe cross-attention below for a given query time-point, $\mathbf{x}_q$ (biases, norms, scaling factor, and linear layer are omitted for brevity):

$$
\begin{aligned}
\text{CrossAttn}(\mathbf{x}_q; \mathbf{X}_k) &= \sum_{\mathbf{x}_k \in \mathbf{X}_k} \frac{\exp(\langle \mathbf{x}_q \mathbf{W}_q, \mathbf{x}_k \mathbf{W}_k \rangle)}{\sum_{\mathbf{x}'_k \in \mathbf{X}'_k} \exp(\langle \mathbf{x}_q \mathbf{W}_k, \mathbf{x}_{k'} \mathbf{W}_k \rangle)} (\mathbf{x}_k \mathbf{W}_v) \\
&= \sum_{\mathbf{x}_k \in \mathbf{X}_k} p(\mathbf{x}_k | \mathbf{x}_q) (\mathbf{x}_k \mathbf{W}_v)
\end{aligned}
\tag{2}
$$

After generalizing $\langle \cdot, \cdot \rangle = \text{sim}(\cdot, \cdot)$ and $\mathbf{x}\mathbf{W} = f(\mathbf{x})$ and reformulating for reconstruction with $\bar{\mathbf{x}}$, we have REBAR and its retrieval formulation in Eq. 3 and Eq. 4, respectively:

$$
\text{REBAR}(\bar{\mathbf{x}}_q; \mathbf{X}_k) = \sum_{\mathbf{x}_k \in \mathbf{X}_k} \frac{\exp(\text{sim}(f_q(\bar{\mathbf{x}}_q), f_k(\mathbf{x}_k)))}{\sum_{\mathbf{x}'_k \in \mathbf{X}'_k} \exp(\text{sim}(f_q(\bar{\mathbf{x}}_q), f_k(\mathbf{x}_{k'})))} f_v(\mathbf{x}_k)
\tag{3}
$$

$$
= \sum_{\mathbf{x}_k \in \mathbf{X}_k} p(\mathbf{x}_k | \bar{\mathbf{x}}_q) f_v(\mathbf{x}_k)
\tag{4}
$$

The retrieval and reconstruction steps of our REBAR($\bar{\mathbf{X}}_q, \mathbf{X}_k$) cross-attention is visualized in Fig. 3. We utilize stacks of dilated convolutions for REBAR's $f_{k/q/v}$, similar to WaveNet (van den Oord et al., 2016), instead of a linear layer. This allows for the retrieval function's similarity function, $\text{sim}(\cdot, \cdot)$, to compare the motifs from the context window around $\bar{\mathbf{x}}_q$ with those in the window around $\mathbf{x}_k$ (Xu et al., 2022). Vanilla cross-attention's linear layer only compares individual time-points with each other. This comparison is illustrated in Fig. 4. The retrieval function, $p(\mathbf{x}_k | \bar{\mathbf{x}}_q)$, identifies regions surrounding an $\mathbf{x}_k \in \mathbf{X}_k$ that are useful for reconstructing $\bar{\mathbf{x}}_q$, and then the $f_v$ consolidates information from that region for reconstruction. See Appendix A.1.1 for further details.

To restate: the objective of REBAR is to learn a class-discriminative measure that will reconstruct well when $\mathbf{X}_k$ has matching motifs with the $\mathbf{X}_q$, but will reconstruct poorly when it does not. Therefore, we would like to *emphasize* a good retrieval function, $p(\mathbf{x}_k | \bar{\mathbf{x}}_q)$ that effectively compares and retrieves motifs and *avoid* a complex model that may achieve an accurate reconstruction even when $\mathbf{X}_q$ and $\mathbf{X}_k$ are dissimilar. *As such, we design our model so that the query is not directly used for reconstruction: it is only used to identify regions in the key subsequence to retrieve with $p(\boldsymbol{x}_k | \bar{\boldsymbol{x}}_q)$.* In other words: for some function $g : \{\Delta^T\}^T \times (T \times D)$,

$$
\text{REBAR}(\bar{\mathbf{X}}_q, \mathbf{X}_k) = g(p(\mathbf{X}_k | \bar{\mathbf{X}}_q), \mathbf{X}_k)
\tag{5}
$$

where $\Delta^T$ is the $T$-dimensional probability simplex: that is, the reconstruction only depends on the query through the probability weights in $p(\mathbf{X}_k | \bar{\mathbf{X}}_q) \in \{\Delta^T\}^T$. The model cannot simply borrow information from within the query to reconstruct itself. For example, if there are 3 time-points on a upwards line and the middle-time point is missing, the model is unable to directly learn a simple linear interpolation to reconstruct that point. Instead, the model is forced to identify a similar upwards line in the key and use this retrieved window for reconstruction. *The model can only reconstruct the query from retrieved regions of the key with $f_v(\boldsymbol{x}_k)$.* As such, reconstruction ability is directly dependent on how similar the motifs in the key are to the query.

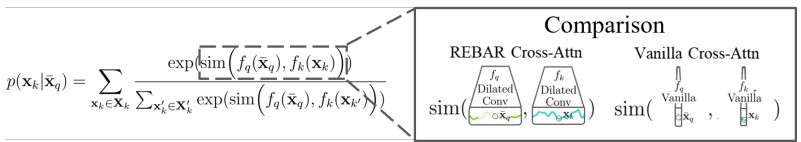

$$p(\mathbf{x}_k|\bar{\mathbf{x}}_q) = \sum_{\mathbf{x}_k \in \mathbf{X}_k} \frac{\exp\left(\text{sim}\left(f_q(\bar{\mathbf{x}}_q), f_k(\mathbf{x}_k)\right)\right)}{\sum_{\mathbf{x}'_k \in \mathbf{X}'_k} \exp\left(\text{sim}\left(f_q(\bar{\mathbf{x}}_q), f_k(\mathbf{x}_{k'})\right)\right)}$$

Figure 4: Comparison of different $f_{q/k}$ within $\text{sim}\left(f_q(\bar{\mathbf{x}}_q), f_k(\mathbf{x}_k)\right)$. The REBAR's $f \coloneqq$ Dilated Convolution allows for semantically-meaningful motif comparison within the retrieval function, unlike in the vanilla's $f \coloneqq \mathbf{xW}$, in which single time-points are compared with another.

Now, we note that training $\text{REBAR}(\bar{\mathbf{X}}_q, \mathbf{X}_k)$ to learn how to retrieve similar motifs for reconstruction is done in the pre-contrastive learning stage, and this should be done without labels so that we can later use REBAR to identify positive pairs for the self-supervised setting. However, given a random $\bar{\mathbf{X}}_q, \mathbf{X}_k$ pair, we do not know if they share class labels, and thus we do not know if reconstruction error should be minimized to learn a motif-matching similarity function between them. Therefore, during training, we set $\mathbf{X}_q$ and $\mathbf{X}_k$ to be the same value, so that REBAR learns a motif similarity function that is able to retrieve the regions from the key $\mathbf{X}_k$ that match the missing region from the query, $\bar{\mathbf{X}}_q$, for reconstruction. Note that we use $\mathbf{X}_q$ and $\mathbf{X}_k$ to indicate two separate variables as inputs to cross-attention, and their values can be the same or different. As previously noted, they are the same during training, but during application, when we use REBAR to identify positive pairs for contrastive learning, $\mathbf{X}_q$ and $\mathbf{X}_k$ are different instances with different values, and REBAR uses its learned motif-retrieval function to reconstruct the query from the most salient motifs in the key. See Appendix A.1.2 for further details on masking methodology during training and application.

## 4.2 APPLYING OUR REBAR MEASURE IN CONTRASTIVE LEARNING

In contrastive learning, the anchor and positive instance are pulled together and the anchor and negative instances are pushed apart in the embedding space. Due to the REBAR measure's aforementioned class-discriminative properties (which is further empirically validated in Sec. 4.3), we can use REBAR to label candidate instances as being positive or negative relative to the anchor.

The trained REBAR cross-attention is used to attempt to reconstruct $\bar{\mathbf{X}}_{\text{anchor}}$ from $\mathbf{X}_{\text{cand}}$. Note that the anchor subsequence $\mathbf{X}_{\text{anchor}}^{(i)}$ and set of candidate subsequences, $\mathcal{S}_{\text{cand}}^{(i)}$, are randomly sampled from the time-series $\mathbf{A}^{(i)}$. Across all of our downstream experiments and datasets, we set $|\mathcal{S}_{\text{cand}}| = 20$. Then, we label the candidates either to be the positive, $\mathbf{X}_{\text{pos}}^{(i)}$, or to be in the within-time-series negative set $\mathcal{S}_{\text{within-neg}}^{(i)}$ based on the reconstruction performance. These labels are then used in our within-time-series loss, $\mathcal{L}_w$, modeled by NT-Xent (Sohn, 2016), shown below.

$$\mathbf{X}_{\text{pos}}^{(i)} \coloneqq \underset{\mathbf{X}_{\text{cand}}^{(i)} \in \mathcal{S}_{\text{cand}}^{(i)}}{\arg\min} \, d(\mathbf{X}_{\text{anchor}}^{(i)}, \mathbf{X}_{\text{cand}}^{(i)})$$

$$\mathcal{S}_{\text{within-neg}}^{(i)} \coloneqq \mathcal{S}_{\text{cand}}^{(i)} \setminus \{\mathbf{X}_{\text{pos}}^{(i)}\}$$

$$\mathcal{L}_w = -\log \frac{\exp(\cos(\mathbf{X}_{\text{anchor}}^{(i)}, \mathbf{X}_{\text{pos}}^{(i)})/\tau)}{\sum_{\mathbf{x}_{\text{neg}}^{(i)} \in \mathcal{S}_{\text{within-neg}}^{(i)}} \exp(\cos(\mathbf{X}_{\text{anchor}}^{(i)}, \mathbf{X}_{\text{neg}}^{(i)})/\tau) + \exp(\cos(\mathbf{X}_{\text{anchor}}^{(i)}, \mathbf{X}_{\text{pos}}^{(i)})/\tau)} \tag{6}$$

This *within-time-series loss, $\mathcal{L}_w$, captures how a time-series can change class labels over time.* It learns to pull the anchor and the subsequence that is most likely to be of the same class as the anchor, according to REBAR, together in the embedding, while pushing those that are less likely, apart.

Next, in order to capture the relationships between-time-series, the other anchor subsequences from the other time-series in our batch, $\mathbf{A}^{(j)}$ with $j \neq i$, are set to be the between-time-series negatives set, $\mathcal{S}_{\text{between-neg}}$. Then, along with our original $\mathbf{X}_{\text{pos}}^{(i)}$, we have our between-time-series loss, $\mathcal{L}_b$.

$$\mathcal{S}_{\text{between-neg}}^{(i)} \coloneqq \bigcup_{j \neq i} \mathbf{X}_{\text{anchor}}^{(j)}$$

$$\mathcal{L}_b = -\log \frac{\exp(\cos(\mathbf{X}_{\text{anchor}}^{(i)}, \mathbf{X}_{\text{pos}}^{(i)})/\tau)}{\sum_{\mathbf{x}_{\text{neg}}^{(j)} \in \mathcal{S}_{\text{between-neg}}^{(i)}} \exp(\cos(\mathbf{X}_{\text{anchor}}^{(i)}, \mathbf{X}_{\text{neg}}^{(j)})/\tau) + \exp(\cos(\mathbf{X}_{\text{anchor}}^{(i)}, \mathbf{X}_{\text{pos}}^{(i)})/\tau)} \tag{7}$$

This *between-time-series loss, $\mathcal{L}_b$, captures differences in time-series, as well as differences in patients*, because commonly, each time-series originates from a different patient. This approach is most similar to augmentation-based methods (e.g. SimCLR) that draw their negatives from the batch.

We utilize a convex combination of these two losses from Eq. 6 and 7 to create our final loss function, $\mathcal{L}$, in Eq. 8, to give us the flexibility between emphasizing learning differences found within-ts or

between-ts. For example, for an accelerometry signals, we could emphasize learning how activity changes over time for a given user by decreasing $\alpha$, and for an ECG signal, we could emphasize how a patient's heart condition is different from other patients' by increasing $\alpha$. See Appendix A.1.3 for further discussion on how $\alpha$ is chosen and its impact on downstream performance.

$$\mathcal{L} = \alpha\mathcal{L}_b + (1-\alpha)\mathcal{L}_w \text{ with } 0 \leq \alpha \leq 1 \tag{8}$$

### 4.3 REBAR NEAREST NEIGHBOR VALIDATION EXPERIMENT

Before using REBAR in contrastive learning, we assess whether the REBAR-identified positive pairs are meaningful, by evaluating whether the REBAR measure effectively predicts mutual class membership. This validation experiment is shown in Eq. 9 and is done by borrowing the class-labels that would typically only be used in downstream experiments. The labels are used in a nearest neighbor classification of an anchor, where distance is measured by REBAR.

$$P(c_{\text{pred}} = c | c_{\text{true}}) = \mathbb{E}_{\mathbf{A}^{(i)} \sim D}\left[\mathbb{E}_{\mathbf{X}^{(i)} \sim \mathbf{A}^{(i)}}\left[\mathbb{1}_{c_{\text{true}}}\left(\underset{c \in \{1, \cdots, C\}}{\arg\min} \, d(\mathbf{X}^{(i)}_{\text{anchor}, c_{\text{true}}}, \mathbf{X}^{(i)}_{\text{cand}, c})\right)\right]\right] \tag{9}$$

This gives us a conditional probability of a predicted class being $c$, given that the anchor is of class $c_{\text{true}}$. One trial randomly segments an anchor subsequence and one candidate subsequence from each of the $C$ classes. This trial is repeated for the given time-series, $\mathbf{A}^{(i)}$, and for all $\mathbf{A}^{(i)}$ in our dataset to obtain each empirical expectation estimate. The specific algorithm details are in Appendix A.4.1. The confusion matrices in Fig. 5 help visualize REBAR's strong results.

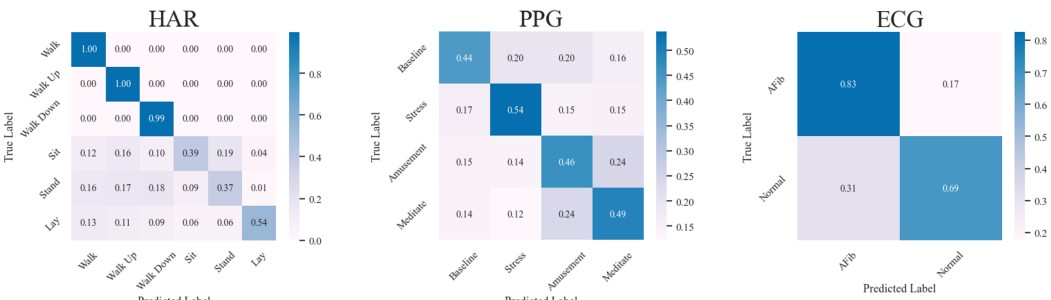

Figure 5: There is a high concentration on the diagonals of the confusion matrices across all of our datasets. This shows that REBAR, although trained with a reconstruction task without class labels, is able to predict mutual class membership, validating our idea for using REBAR to identify positive pairs in contrastive learning.

Our three datasets are further explained in the Sec. 5 and in Appendix A.2, but what is most important to note is that each of them represent distinctively different sensor modalities. Given this, across all three distinctive domains, each true label's highest prediction label is still always itself, such that $c_{\text{true}} = \arg\max_c P(c_{\text{pred}} = c | c_{\text{true}})$. This implies that REBAR-identified positive pairs will match the anchor with its correct class more often than any of the other individual classes, and so using REBAR to train our contrastive learning framework will encourage mutual classes to be pushed together in the embedding space over time. This validates our usage of REBAR in the unsupervised setting. As previously noted in Sec. 4.2, we use REBAR to compare an anchor subsequence with a set of randomly sampled candidate subsequences in order to identify the positive candidate.

## 5 DOWNSTREAM EXPERIMENTS AND RESULTS

In this section, we detail our experimental design for evaluating REBAR against other contrastive learning methods, and the results with an ablation study are shown in Sec 5.1.

**Benchmarks:** We aim to assess how differing methods perform based upon their contrastive objective, so the encoder architecture (Yue et al., 2022) is kept constant across all benchmarks. Each benchmark represents a specific time-series contrastive learning paradigm, and they are listed below. Further implementation details are found in Appendix A.3.

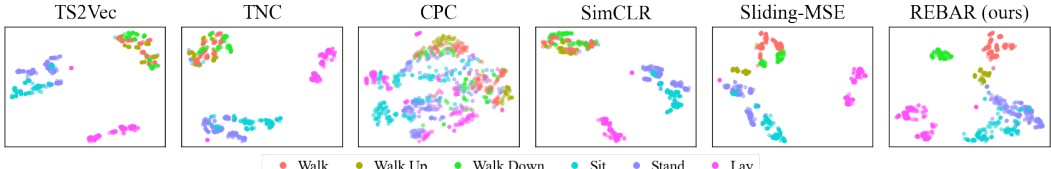

Figure 6: Qualitative Clusterability results with t-SNE Visualizations of each benchmark's encodings on the HAR dataset. Most methods are able to encode "Lay" into its own cluster, and also "Sit" and "Stand" into nearby but generally distinct clusters. REBAR is the only method able to successfully separate out the "Walk", "Walk Up", and "Walk Down" labels into disjoint clusters.

- *TS2Vec* is a strong augmentation-based method with time-stamp representations (Yue et al., 2022).
- *TNC* is a sampling-based method that samples a nearby subsequence as a positive and utilizes a hyperparameter to estimate whether a distant sample is negative (Tonekaboni et al., 2021).
- *CPC* contrasts based on future timepoint predictions (Oord et al., 2018).
- *SimCLR* is a simple augmentation-based method (Chen et al., 2020), which we have adapted for time-series with the most common augmentations (i.e. shifting, scaling, jittering).
- *Sliding-MSE* is a simplified REBAR that instead uses a sliding-MSE comparison as a measure.

**Data:** We utilize 3 datasets from 3 different sensor domains with time-series that have their classification labels change over time: Human Activity Recognition (HAR) with accelerometer and gyroscopic sensors to measure activity (Reyes-Ortiz et al., 2015), PPG to measure stress (Schmidt et al., 2018), and ECG to measure heart condition (Moody, 1983). Each of these modalities have drastically different structures, and the class-specific temporal patterns vary differently within a modality. Please find further dataset descriptions and visualizations in Appendix A.2.

**Downstream Evaluation:** To evaluate their class-discriminative strengths, we learn a linear probe (i.e. logistic regression) on each model's frozen encoding for downstream classification and use the Accuracy, AUROC, and AUPRC metrics to quantify the results. Additionally, a fully supervised model composed of an encoder, identical to that used by the baselines, with a linear classification layer is benchmarked. This matches our baselines' linear probe evaluation, only trained end-to-end. We then also assess cluster agreement for further corroboration. After a $k$-means clustering of the frozen encoding, with $k$ as the number of classes, we assess the similarity of these clusters with the true labels with the Adjusted Rand Index and Normalized Mutual Information metrics.

## 5.1 RESULTS

**Linear Probe Classification:** Tbl. 1 shows that the linear probe trained on our REBAR representation consistently achieved the strongest results, even beating the fully supervised model in PPG and HAR, achieving the same accuracies, but higher AUROC and AUPRC. For ECG, REBAR achieves better accuracy, but lower AUROC and AUPRC. This demonstrates our REBAR's methods strength in learning a representation that is better at handling class imbalance than the fully-supervised model.

| | HAR | | | PPG | | | ECG | | |
|---|---|---|---|---|---|---|---|---|---|
| Model | ↑ Accuracy | ↑ AUROC | ↑ AUPRC | ↑ Accuracy | ↑ AUROC | ↑ AUPRC | ↑ Accuracy | ↑ AUROC | ↑ AUPRC |
| Fully Supervised | **0.9535** | 0.9835 | 0.9531 | **0.4138** | 0.6241 | 0.3689 | 0.7814 | **0.9329** | **0.9260** |
| TS2Vec | 0.9324 | 0.9931 | 0.9766 | 0.4023 | 0.6428 | 0.3959 | 0.7612 | 0.8656 | 0.8516 |
| TNC | 0.9437 | 0.9937 | 0.9788 | 0.2989 | 0.6253 | 0.3730 | 0.7340 | 0.8405 | 0.8195 |
| CPC | 0.8662 | 0.9867 | 0.9438 | 0.3448 | 0.5843 | 0.3642 | 0.7775 | 0.8377 | 0.8223 |
| SimCLR | 0.9465 | 0.9938 | 0.9763 | 0.3448 | 0.6119 | 0.3608 | 0.6992 | 0.8254 | 0.8063 |
| Sliding-MSE | 0.9352 | 0.9931 | 0.9767 | 0.3333 | 0.6456 | 0.3831 | 0.7751 | 0.8755 | 0.8574 |
| REBAR (ours) | **0.9535** | **0.9965** | **0.9891** | **0.4138** | **0.6977** | **0.4457** | **0.8154** | 0.9146 | 0.8985 |

Table 1: Linear Probe Classification Results with Accuracy, AUROC, and AUPRC

Our REBAR method demonstrates a much stronger performance than Sliding-MSE, showing the necessity of learning a retrieval-reconstruction distance rather than using a simple measure to identify positive pairs. Our improved performance compared to TNC highlights the value of identifying positives that are not necessarily near the anchor. SimCLR's subpar results demonstrate that even if common time-series augmentations are used, this does not guarantee strong performance and the set of augmentations should be tuned. Although TS2Vec is a state-of-the-art method, it is unable to consistently achieve the strongest performance among the other benchmarks. We suspect that

this is because TS2Vec was evaluated on short class-labeled time-series rather than time-series with class-labeled subsequences. REBAR's sampling-based approach successfully exploits this structure to sample positive pairs from subsequences across the time to achieve strong performance.

**Clusterability Evaluation:** Tbl. 2 shows that when measuring the cluster agreement with the true class labels, REBAR continues to achieve the best ARI and NMI, corroborating the strong classification results. This is unlike other methods, such as TS2vec in PPG, that achieve strong linear probe results, but low cluster agreement. Additionally, the t-SNE visualizations shown in Fig. 6 for HAR and in Appendix A.4.2 for remaining datasets show that REBAR's encodings push mutual classes together and distinct classes apart, even distinct classes that are semantically similar, such as "Walk", "Walk Down", and "Walk Up". These results support the idea that REBAR's embedding space is particularly class-discriminative.

| | HAR | | PPG | | ECG | |
|---|---|---|---|---|---|---|
| Model | ↑ ARI | ↑ NMI | ↑ ARI | ↑ NMI | ↑ ARI | ↑ NMI |
| TS2Vec | 0.4654 | 0.6115 | -0.0353 | 0.1582 | 0.2087 | 0.1701 |
| TNC | 0.4517 | 0.5872 | 0.0958 | 0.1666 | 0.2186 | 0.1753 |
| CPC | 0.1603 | 0.2217 | 0.1110 | 0.1867 | 0.0532 | 0.0724 |
| SimCLR | 0.5805 | 0.6801 | 0.1535 | 0.3081 | 0.2182 | 0.1751 |
| Sliding-MSE | 0.5985 | 0.7019 | 0.1083 | 0.2141 | -0.0081 | 0.0180 |
| REBAR (ours) | **0.6258** | **0.7721** | **0.1830** | **0.3422** | **0.2260** | **0.1796** |

Table 2: Clusterability Results with Adjusted Rand Index and Normalized Mutual Information

**REBAR approach analysis:** Fig. 7 visualizes the positive pairing that was identified by our REBAR measure from a randomly sampled set of candidates for a given anchor for the HAR dataset. We see that even when there is no exact match of the anchor within the candidates, REBAR's motif-comparison retrieval and reconstruction is able to identify a positive example that shares the same class as the anchor. Please find a large gallery of positive pairing visualizations in Appendix A.4.4.

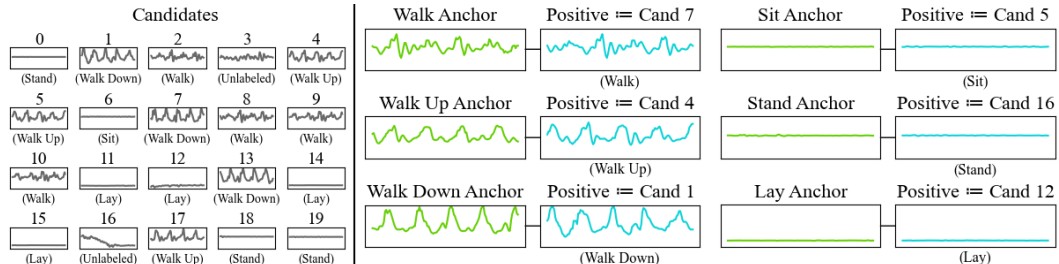

Figure 7: Positive pairings identified by REBAR from the candidates, for an anchor of each class.

The key component of our REBAR cross-attention design is the dilated convolutions used for motif comparison and retrieval. We find that removing and replacing these convolutions with the vanilla linear layer, results in performance drops of a 6.4% decrease in accuracy of the linear probe, which is worse than all but one of the benchmarks, and a 34.1% NMI decrease in the clusterability evaluation. Additionally, we find that REBAR is fairly robust against hyperparameter tuning. When we modify the size of the dilated conv's receptive field, the size of masks, or REBAR cross-attention reconstruction training epochs, performance remains consistent. See Appendix A.1.4 for further details and additional specific model ablation results.

## 6 CONCLUSION

In this paper we introduced REBAR, a novel approach to time-series contrastive learning. By using cross-attention to retrieve class-specific motifs in one subsequence to reconstruct another subsequence, we can predict mutual class membership. Then, if we use this REBAR measure to identify positive pairs, we are able to achieve state-of-the-art results in learning a class discriminative embedding space. Our REBAR method offers a new perspective in time-series self-supervised learning with our measure-focused approach, and we hope that this work will drive future research into how to best capture and encode semantic relationships between time-series.

## 7 Acknowledgements

We would like to thank Catherine Liu for her help and support on this work. This work is supported in part by NIH P41-EB028242-01A1, NIH 1-R01-CA224537-01, and the National Science Foundation Graduate Research Fellowship under Grant No. DGE-2039655. Any opinion, findings, and conclusions or recommendations expressed in this material are those of the authors and do not necessarily reflect the views of the National Science Foundation.

## 8 Ethics Statement

Our paper works on creating models for health-related signals, and it has the potential to improve health outcomes, but at the same time could lead to a loss of privacy and could possibly increase health-related disparities by allowing providers to characterize patients in more fine-grained ways. In the absence of effective legislation and regulation, patients may lack control over use of their data, leading to questions of whether autonomy, a key pillar of medical ethics, are being upheld. Overall though, we hope that our work leads to a net positive as it helps further the field towards creating personalized health recommendations, allowing patients to receive improved care and achieve better health outcomes, directly contributing to patient safety and overall well-being.

## 9 Reproducibility Statement

Our Methods section in Section 4 details the way in which we set-up our method, and our Experiments section in Section 5 details our experimental design. Additionally, in the Appendix A.3, we itemize each of the hyperparameters we used to tune each of our benchmarks. Upon acceptance, we will release our GitHub code publicly, which will have the set seeds and exact code we used to run our experiments. We will also make our model checkpoints downloadable. The datasets used are publicly available, and we describe how we curate each of them for our task in Appendix A.2. Additionally, our code can be found at `https://github.com/maxxu05/rebar`.

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

# A APPENDIX

## A.1 REBAR APPROACH DETAILS

We use this section of the appendix to explain and justify the architectural and implementation details of the full REBAR approach. In Sec. A.1.1, we further explain the specific REBAR cross-attention architecture details, including the dilated convolution block used for motif comparison and retrieval. In Sec. A.1.2, we explain the masking procedure we use during training and application of our REBAR cross-attention. In Sec. A.1.3, we explain how we choose an $\alpha$ in our combined loss function, emphasizing within-time-series interactions and/or between-time-series interactions, and its effects on downstream performance. In Sec. A.1.4, we detail our ablation study and parameter robustness analysis.

### A.1.1 REBAR CROSS-ATTENTION ARCHITECTURE DETAILS

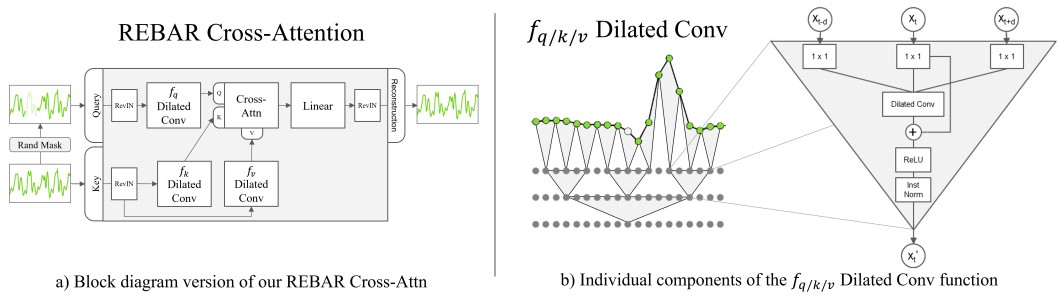

a) Block diagram version of our REBAR Cross-Attn      b) Individual components of the $f_{q/k/v}$ Dilated Conv function

Figure 8: REBAR Cross-Attention Architecture

Fig. 8a) contains the full block diagram version of the REBAR cross-attention originally shown in Fig. 3, with all of the model details included, such as the RevIN block used for normalization and the aggregation linear layer, used to combine information across heads for final reconstruction. Below we explain each of the components and the rationale behind including/excluding them.

- **Dilated Convolution**: We use a modified version of the BDC transformer proposed in PulseImpute (Xu et al., 2022), which is visualized in Fig. 8b). The instance norm (Ulyanov et al., 2016) is used to normalize the inputs, and an instance norm is used instead of a batch norm so that the model is not dependent on batch size. The 1x1 layer acts as a bottleneck layer, reducing dimensionality. This allows for the model to be able to efficiently stack convolution layers with exponentially increasing dilation factors, thereby exponentially and efficiently increasing the receptive field. The residual connections are designed to help to facilitate more efficient training. Please see in the Sec. 4.1 in the main text for explanation of the importance of using a dilated convolution for $f_{q/k/v}$ rather than vanilla cross-attention's linear layer.
- **Positional Encoding Exclusion**: Positional encoding was proposed in the original transformer model (Vaswani et al., 2017) in order to address the permutation equivariance problem of self-attention, so that the model would have positional awareness of each of the input tokens. However, as noted in PulseImpute (Xu et al., 2022), the addition of a dilated convolution removes the permutation equivariance limitation. We opt to not use any positional encoding because we would like the retrieval function, $p(\mathbf{x}_k|\bar{\mathbf{x}}_q)$, to retrieve regions in the key subsequence based on similarity of the temporal shapes within the key and query in order to assess motif-similarity, rather than considering the specific positions of such structures. Additionally, knowing the exact positions is uninformative due to the arbitrary time at which sensors begin recording or are segmented. Thus, we seek to instead model relative position through the convolutions that can capture a subsequence's shape.
- **Partial Convolution**: To handle masking, we do a drop-in replacement of all convolutions with their partial convolution counterpart (Liu et al., 2018). Partial convolutions ignore the calculation at masked points and reweigh the weights based on the total number of weights that were used in the calculation.
- **RevIN**: Reversible instance norm (RevIN) was originally proposed to help address temporal distribution shifts in forecasting tasks, and it uses an ordinary instance norm at inputs and then reverses the normalization at the output (Kim et al., 2021). We use RevIN normalize the query and key

inputs with respect to the query, then is reversed at the end to unnormalize the outputs for reconstruction.

### A.1.2 REBAR Masking Procedure

We must identify an effective masking strategy to train $\text{REBAR}(\bar{\mathbf{X}}_q, \mathbf{X}_k)$ to learn to retrieve and compare class-specific motifs (e.g. heartbeat in ECG (Schäfer & Leser, 2022)) and also to apply $\text{REBAR}(\bar{\mathbf{X}}_{\text{anchor}}, \mathbf{X}_{\text{cand}})$ to effectively identify meaningful positive and negative examples from the candidates. We define our binary mask as $m$ and two different types of specific masks, *extended* and *transient*, below.

$$m = \begin{cases} 1 & \text{for missing} \\ 0 & \text{for not-missing} \end{cases}, \text{where } m \in \mathbb{R}^T$$

$$m_{\text{extended}}[i : i + n] = 1 \text{ with } i \in_R \{1, \cdots, T\}, 0 \text{ else.}$$

$$m_{\text{transient}}[i_1, \cdots, i_n] = 1 \text{ with } i_1, \cdots, i_n \in_R \{1, \cdots, T\}, 0 \text{ else.}$$

An extended mask is where a contiguous segment of length $n$ is randomly masked out, and a transient mask is where $n$ time stamps are randomly masked out. $\in_R$ designates random sampling without replacement.

**Training REBAR$(\bar{\mathbf{X}}_q, \mathbf{X}_k)$ before Contrastive Learning:** The key idea is that we want learn REBAR such that the reconstruction error is class-discriminative, and the model's reconstruction ability direction depends on the retrieval function.

Fig. 9b) shows that training with the transient mask results in a dispersed attention distribution, which means that the model is retrieving minor, non-unique motifs. Even if two $\mathbf{X}_k$ have two different class labels, we would be able to retrieve this non-unique motif from both $\mathbf{X}_k$, so that the reconstruction performance would not be class-discriminative.

Fig. 9a) shows that the attention weights learned with a extended mask fit exactly onto the region in the key corresponding to the masked region in the query, and thus the model has learned to retrieve specific motifs useful for reconstruction. Because the REBAR model has learned to compare specific motifs between the query and key, when comparing class-discriminative motifs, the learned reconstruction performance will be class-discriminative.

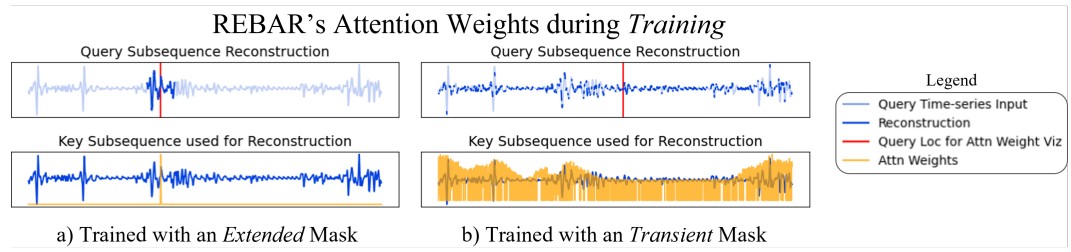

a) Trained with an *Extended* Mask     b) Trained with an *Transient* Mask

Figure 9: a) A extended mask encourages REBAR to learn attention weights that are specific to the corresponding motif in the key. Conversely, with an transient mask, b), the attention is diffused and attends to transient motifs throughout the key. Specific attention weights allow for an effective comparison of class-discriminative motifs, so we utilize a extended mask for training.

**Applying REBAR$(\bar{\mathbf{X}}_{\text{anchor}}, \mathbf{X}_{\text{cand}})$ for Contrastive Learning:** During application, the REBAR cross-attention has already been trained to capture potentially class-disciriminative motifs, and we use the REBAR reconstruction error to identify pairs that are likely to share a class, so that they can be labeled as positive examples. Then, when these positive pairs are pulled together within the embedding space, they encode class-discriminative information. Therefore, during application, we want to utilize the trained REBAR cross-attention in a way that holistically compares one subsequence with another, so that we have a better understanding whether they share mutual classes. As noted in Equation 5, we know that $p(\mathbf{X}_k | \bar{\mathbf{X}}_q)$ is the only place in which the query, $\mathbf{X}_q$, is used for reconstruction. Thus, we look at visualizations of $p(\mathbf{X}_k | \bar{\mathbf{X}}_q)$ for an extended mask vs. a transient mask, in Fig. 11, to understand what motifs within the query are compared to the key, during reconstruction.

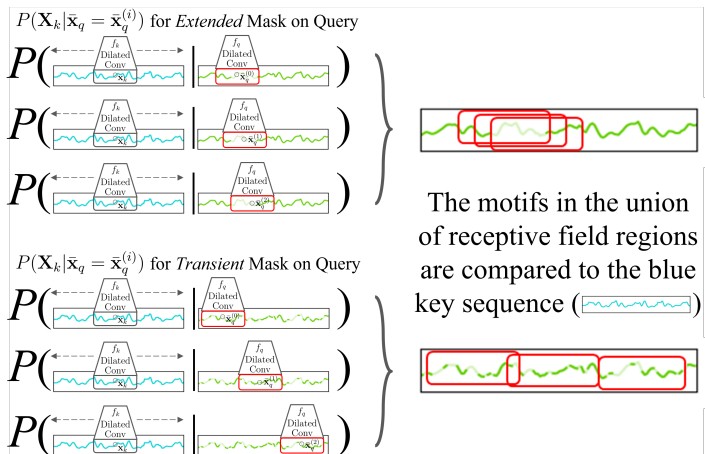

Figure 10: Comparison of receptive fields of the query being considered in the query key motif comparison found in $p(\mathbf{X}_k|\bar{\mathbf{x}}_q)$. We can see that the union of receptive fields of the $f_q$ Dilated Convolution covers more area when reconstruction is done over a transient mask, rather than an extended mask.

At application time, a extended mask could be used, however, in doing so, only the motifs near the contiguously masked out region would be compared to the key. This is because when reconstructing a given masked time-point and another point contiguous to it, the receptive fields to be used to identify motifs to be compared to the key are heavily overlapping, see the top of Fig. 10.

A transient mask allows for many different motifs in the query, each of them captured in a receptive field around the many masked time-points dispersed throughout the signal, to be compared to the key during reconstruction. This allows for a higher coverage of the query in conducting such motif-similarity comparisons with the key. Therefore, during application, when we are testing each candidate as the key for a given anchor as the query, we would be able to identify the candidate that is most similar to the anchor as a whole. See the bottom of Fig. 10.

A transient mask preserves more information about the surrounding context of a masked out query point compared to an extended mask, and we see that a model trained with a extended mask can still reconstruct a signal masked with a transient mask, as seen in Fig. 11. The vice versa would likely not be true and is reflected in the empirical results shown in Table 3. Even though the masking setting has changed, each of the reconstructions still maintains a sparse attention, which implies the retrieval function is still specific and comparing specific, potentially class-discriminative, motifs between the query and the key.

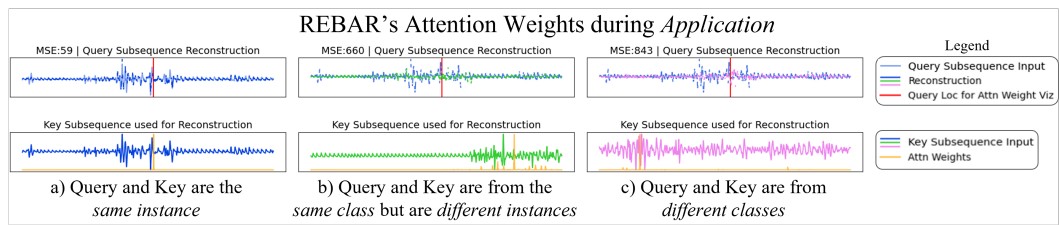

Figure 11: We use a transient mask during application to evaluate motif similarity on a wider range of motifs present throughout the query, and the sparse attention in c) d) e) show that the model is still conducting specific motif comparison. c) and d) utilize keys that differ from the query, but our REBAR cross-attention still retrieves the most relevant information in the key for query reconstruction. The attention weights are conditioned on a masked-out downwards dip in the query, and each of the weights in c) d) e) are highest for a corresponding dip in the key. d)'s key is in the same class as the query, and the corresponding MSE is lower than e)'s.

**Masking Procedure Empirical Validation Experiments**: As we see in Table 3, Training Extended + Application Transient achieves the best performance in our validation experiment, followed by Training Extended + Application Extended, then Training Transient + Application Transient, then Training w/ Transient + Application Extended.

| Training Mask Type Evaluation Mask Type | Extended Extended | Extended Intermittient | Intermittient Intermittient | Intermittient Extended |
|---|---|---|---|---|
| Accuracy | 0.3624 | **0.4580** | 0.2888 | 0.2378 |

Table 3: Comparison of Different Masking Procedures for REBAR. The accuracy reported is from the validation strategy explained in Section 4.3 averaged across all classes.

### A.1.3 EMPHASIZING WITHIN-TIME-SERIES INTERACTIONS VS. BETWEEN-TIME-SERIES INTERACTIONS IN OUR LOSS FUNCTION

The $\alpha$ coefficient in our loss function in Equation 8 gives us the flexibility to learn an embedding space that emphasizes how a time-series changes over time when decreasing $\alpha$ vs. how a time-series differs across patients when increasing $\alpha$. We hypothesize the best $\alpha$ for each modality.

- For ECG, these signals measure the electrical propagation throughout the heart, and each patient's ECG signals is very specific to their own heart and how a specific cardiac condition, such as Atrial Fibrillation, affects it. There is even research showing that ECG signals could be used to potentially identify an individual due to the unique position, shape, size, and structure of the heart (Lee et al., 2018b). Therefore, for ECG, it would make more sense to utilize a loss function that emphasizes differences among patients with an $\alpha = 1$.
- For HAR, as we see in the time-series visualizations in Appendix A.2, each of the different activities accelerometry signals are quite distinct from each other within a given user's time-series, so learning how a time-series changes over time with an $\alpha = 0$ is the most useful for distinguishing classes.
- For PPG, this signal modality shares similar properties from both ECG and HAR. Because PPG measures blood volume changes, it models unique properties of how a user's heart pumps blood, but it also is important in how it models how a specific user responds to stress. Therefore, we use a combination of the two loss functions with an $\alpha = 0.5$.

For each of the datasets, we benchmark three different values of $\alpha \in \{0, 0.5, 1\}$ for having only the within-time-series loss function, both loss functions, or only the between-time-series loss function, respectively. Please see the results below in Table 4.

| | HAR | | | | | PPG | | | | | ECG | | | | |
|---|---|---|---|---|---|---|---|---|---|---|---|---|---|---|---|
| | Accuracy | AUROC | AUPRC | ARI | NMI | Accuracy | AUROC | AUPRC | ARI | NMI | Accuracy | AUROC | AUPRC | ARI | NMI |
| $\alpha$=0 | **0.9535** | **0.9965** | **0.9891** | **0.6258** | **0.7721** | 0.4713 | 0.7215 | 0.4528 | 0.0408 | 0.1163 | 0.8298 | 0.8946 | 0.8775 | 0.1995 | 0.1470 |
| $\alpha$=.5 | 0.9549 | 0.9941 | 0.9802 | 0.4787 | 0.6273 | **0.4138** | **0.6977** | **0.4457** | **0.1830** | **0.3422** | 0.7944 | 0.9339 | 0.9286 | 0.2263 | 0.1800 |
| $\alpha$=1 | 0.9056 | 0.9898 | 0.9668 | 0.3077 | 0.4430 | 0.4138 | 0.6398 | 0.3816 | 0.0586 | 0.2242 | **0.8154** | **0.9146** | **0.8985** | **0.2260** | **0.1796** |

Table 4: Benchmarking REBAR model with different values of $\alpha$ within our loss function in Eq. 8. $\alpha$=0 is only the within-time-series loss function, $\alpha$=1 is only the between-time-series loss function, and $\alpha$=.5 is an equal contribution between the two of them. Bold is the variant that achieved the best balance of high linear probe performance and high clusterability, and it is what we reported within our results in the main text, Sec. 5.1.

REBAR achieves strong performance under all loss function variants, but switching between $\alpha$ values results in different trade-offs. For example, in ECG, the model we end up reporting in our results uses an $\alpha = 1$, and it achieves an accuracy of .8154 and NMI of .1796, but if we use $\alpha = 0$, we can achieve a higher accuracy of .8946, but a lower NMI of 0.1470. Then, in PPG, when $\alpha = 0$, then we achieve the highest accuracy of 0.4713, but the NMI is only 0.1163, which is much lower than $\alpha = 0.5$, where NMI is 0.3422. In our main results, we utilize the $\alpha$ that provides the best balance between high linear probe performance and high clusterability, and these chosen $\alpha$ values align with our prior hypotheses.

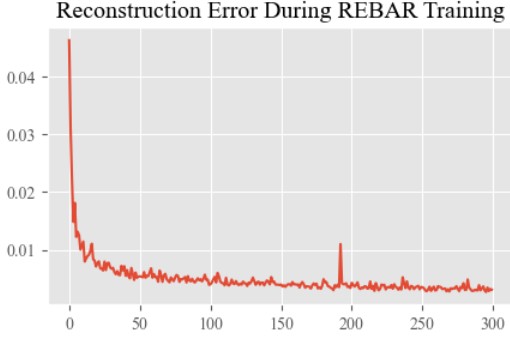

Figure 12: As the REBAR Cross-Attention trains with the extended mask, reconstruction error decreases, but reconstruction error is not necessarily an indicator of downstream performance.

### A.1.4 REBAR CROSS-ATTENTION ABLATION STUDY AND HYPERPARAMETER ANALYSIS

| | Acc | AUROC | AUPRC | ARI | NMI |
|---|---|---|---|---|---|
| | Baseline at Epoch 300, Mask=15, RF=43 | | | | |
| REBAR | 0.9535 | 0.9965 | 0.9891 | 0.6258 | 0.7721 |
| | Ablation (Lower is better) | | | | |
| w/o DilatedConv | 0.8930 (-0.061) | 0.9839 (-.0130) | 0.9302 (-0.059) | 0.2845 (-0.341) | 0.5094 (-0.263) |
| w/o RevIN | 0.9451 (-0.008) | 0.9958 (-0.001) | 0.9861 (-0.003) | 0.5613 (-0.065) | 0.6450 (-0.127) |
| w/ PosEmbed | 0.9479 (-0.006) | 0.9957 (-0.001) | 0.9863 (-0.003) | 0.5550 (-0.071) | 0.6522 (-0.120) |
| | Robustness (Equal or Higher is better) | | | | |
| REBAR at Epoch 100 | 0.9592 (+0.006) | 0.9970 (+0.000) | 0.9900 (+0.001) | 0.6169 (-0.009) | 0.7589 (-0.013) |
| REBAR at Epoch 200 | 0.9592 (+0.006) | 0.9967 (+0.000) | 0.9896 (+0.001) | 0.6830 (+0.057) | 0.8425 (+0.070) |
| REBAR w/ Mask=5 | 0.9493 (-0.004) | 0.9963 (+0.000) | 0.9881 (-0.001) | 0.6120 (-0.014) | 0.7550 (-0.017) |
| REBAR w/ Mask=25 | 0.9592 (+0.006) | 0.9962 (+0.000) | 0.9873 (-0.002) | 0.6484 (+0.023) | 0.7891 (+0.017) |
| REBAR w/ RF=31 | 0.9507 (-0.003) | 0.9962 (+0.000) | 0.9881 (-0.001) | 0.6390 (+0.013) | 0.7787 (+0.007) |
| REBAR w/ RF=55 | 0.9380 (-0.016) | 0.9943 (-0.002) | 0.9794 (-0.010) | 0.6570 (+0.031) | 0.8090 (+0.037) |
| | Failure Cases (Lower is better) | | | | |
| REBAR at Epoch 1 | 0.9437 (-0.010) | 0.9954 (-0.001) | 0.9841 (-0.005) | 0.3635 (-0.290) | 0.5213 (-0.251) |

Table 5: REBAR model Ablation Study and Hyperparameter Analysis on HAR data

**Ablation Study of Model Components**: Tbl. 5 demonstrates that each of the REBAR cross-attention model's design contributes to its strong contrastive learning performance, especially the dilated convolutions (e.g. .061 drop in accuracy and a .263 drop in NMI). This makes sense because the dilated convolutions enable the motif comparison and retrieval. The exclusion of our reversible instance norm leads to a .008 drop in accuracy and a .127 drop in NMI, and the addition of an explicit positional embedding leads to a .006 drop in accuracy and .120 drop in NMI. As noted in Section 3.1, we intentionally keep the design of REBAR attention simple to emphasize the motif comparison within the cross-attention mechanism.

**Effect of Proportion of Masking**: The model is fairly robust against the size of the extended masks to be used for training the REBAR model. In our ablation study in Tbl. 5, compared to an initial mask size of 15, we found that using a smaller mask size of 5 time-points lowered accuracy by only 0.004 and using a larger mask size of 25 increased accuracy by .006.

For each of the datasets, the size of the extended mask is chosen to be large enough to mask out a potentially class-discriminative motif. This is done so that the model learns how to specifically retrieve this class-discriminative motif from the key and so that during evaluation, class-discriminative motifs between the key and query are compared. However, the mask size cannot be too large, because then even if the query and key are of the same class, it would be more difficult to precisely match a much larger motif with the increased noise.

Thus, extended mask size can be tuned with a hyperparameter grid search or with expert domain knowledge. In HAR, it is a mask that covers .3 seconds (15 time-points) because the accelerometry type data typically captures information of movement from short quick jerks. In PPG, it is a mask that covers 4.69 seconds (300 time points), which we do because PPG wave morphology is relatively simple, so we would like to capture a few quasi-periods. In ECG it is a mask that covers 1.2 seconds (300 time points), which covers a heartbeat, and ECG wave morphology is generally very descriptive and specific (e.g. QRS complex).

**Effect of Dilated Convolution Receptive Field Size**: This also helps guide the size of the receptive field that our dilated convolutions in our cross-attention should be. In general, we aim for the size of the receptive field to be three times the size of the extended mask, so that during training, the query dilated convolution would be able to effectively capture the motif surrounding the masked out query region to learn our cross-attention model. Additionally, as we see in the Tbl. 5, our model is robust to the exact receptive field size. Compared to an initial receptive field of 43, a receptive field of 31 decreases accuracy by only .003 while increasing ARI by .013 and a receptive field of 55 decreases accuracy by only .016 while increasing ARI by .031.

**Effect of Reconstruction on Downstream Performance**: Tbl. 5 shows that REBAR is robust to the hyperparameter of Number of REBAR Cross-Attention Epochs Trained. Fig. 12 shows that reconstruction error decreases as epochs trained increases, so reconstruction accuracy is not necessarily an indicator of downstream performance. This is not an issue because the REBAR Cross-Attention's goal within a contrastive learning framework is not to achieve a good reconstruction. Instead, we would like the REBAR Cross-Attention to learn a good retrieval and motif-comparison function between the query and the key that achieves better or worse reconstruction, based on motif similarity. As noted in Section 3.1, we intentionally keep the design of REBAR attention simple to emphasize the motif comparison within the cross-attention mechanism. We note that these results should not imply that the REBAR model should not be trained. Tbl. 5 in the Appendix shows that if REBAR is only trained for 1 epoch, then accuracy drops by 0.010 and NMI drops by 0.251.

### A.1.5 COMPARING REBAR TO MASKED AUTOENCODER

The Masked Autoencoder (MAE) has been primarily used in NLP (Devlin et al., 2018) and Vision (He et al., 2022), with a few works in time-series (Cheng et al., 2023; Li et al., 2023). These models typically utilize transformers to encode the masked input before reconstructing with a lightweight decoder. While both REBAR and MAEs involve masked reconstruction, the fundamental approaches differ substantially. REBAR estimates if a pair of subsequences are of the same class via a paired reconstruction error and then uses this to contrast subsequences. In comparison, MAEs learn between-time-point interactions within a subsequence by learning a reconstruction as a composition of the unmasked data and is not designed to explicitly compare subsequences.

### A.2 DATASET DETAILS

The time-series are split into a 70/15/15 train/val/test split. We note below that the total time coverage of the class-labeled subsequences does not equal to the total time coverage of the time-series that they are segmented from, because some of the datasets contain parts that were unlabeled. Additionally, each of the class-labeled subsequences are non-overlapping.

**HAR**: Rather than using the extracted time and frequency features, we opt to use the raw accelerometer and gyroscopic sensor data (Reyes-Ortiz et al., 2015) to better assess how our methods perform on raw time-series. Our subsequences are 2.56-second-long (128 time points) to match the subsequence length proposed for classification in the original work (Reyes-Ortiz et al., 2015). There are 6,914 distinct subsequences total for training and testing our models, which are sampled from 59 5-minute-long (15,000) time-series that are sampled at 50 Hz and has 6 channels. There are 4,600 class-labeled subsequences, with the following labels: walking (17.7%), walking upstairs (7.6%), walking downstairs (9.1%), sitting (18.2%), standing (20.1%), and laying (20.1%).

**PPG**: From the WESAD dataset (Schmidt et al., 2018), we use 1-minute-long (3840 time points) subsequences to match the subsequence length proposed in the original work (Schmidt et al., 2018). There are 1,305 distinct subsequences total for training and testing our models, which are sampled from 15 87-minute-long (334,080 time points) time-series, sampled at 64 Hz with a single channel. There are a total of 666 class-labeled subsequences, with the following labels: the labels are baseline

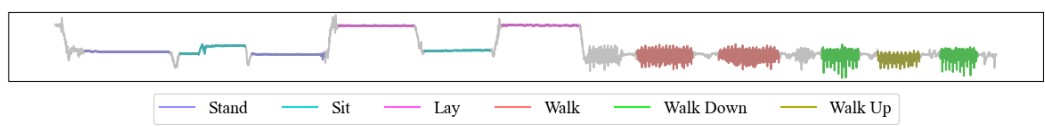

Figure 13: Example of one channel of HAR Signal

(42.7%), stress (24.0%), amusement (12.4%), and meditation (20.9%). The data has been denoised following the procedure in Heo et al. (2021).

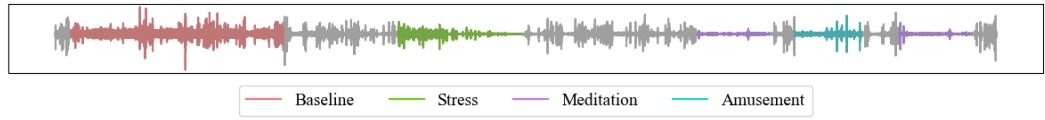

Figure 14: Example of PPG Signal

**ECG**: From the MIT-BIH Atrial Fibrillation dataset (Moody, 1983), we use 10-second-long (2500 time points) subsequences. The original work (Moody, 1983) does not have a proposed subsequence length, so we use this 10-second-long subsequence length to match the prior work for this dataset (Tonekaboni et al., 2021) and the 10-second-long length used in general ECG classification works (Wagner et al., 2020). There are 76,590 distinct subsequences total for training and testing our models, which are sampled from 23 9.25-hour-long (8,325,000 time points) time-series sampled at 250 Hz with dual channels. There is a total of 76,567 10-second-long (2500 time points) class-labeled subsequences, with the following labels: atrial fibrillation (41.7%) and normal (58.3%). These subsequences originate from 23 9.25-hour-long (8,325,000 time points) time-series sampled at 250 Hz with dual channels.

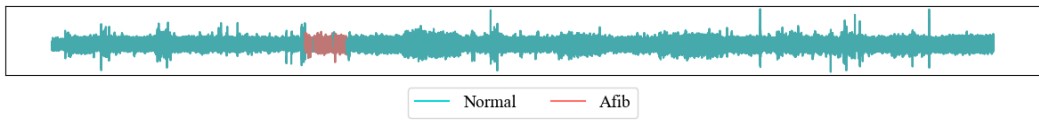

Figure 15: Example of one channel of ECG Signal

### A.2.1 DISCUSSION ABOUT UCR/UEA TIME-SERIES DATABASE

We note that we opt to not benchmark on the commonly utilized UCR and UEA time-series dataset databases, as 1) they do not fit our stated problem area of time-series that are series of class-labeled subsequences, 2) would prevent us from benchmarking sampling-based contrastive learning methods, and 3) many of datasets found within the databases are very small.

Our work is framed for long time-series data that can be described as a series of class-labeled subsequences, rather than a collection of short class-labeled time-series (e.g. UCR, UEA). For context, our datasets have time-series that are 8,325,000 / 334,080 / 15,000 time points long, compared to the median length of 621 and 301 for UEA and UCR respectively. Our defined subset of time-series data is the closest primitive to true time-series sensors, in which sensors are designed to collect large volumes of unlabeled data continuously and autonomously. We note that this setting particularly describes the physiological sensor time-series domain, in which the advancement of wearable technologies has enabled large data collection on a user's changing physiological states, but the cost of labeling remains high (i.e. mHealth sensors require constant user annotation and clinical sensors require medical expertise). Hence, we believe that this particular time-series domain has the largest opportunity for impact in the ML and greater health community.

Using this type of long time-series data in our problem also allows us to benchmark sampling-based contrastive learning methods, such as TNC (Tonekaboni et al., 2021) and our own REBAR method. These sampling-based methods fully exploit the time-series nature of our task. Because each subsequence is part of a specific user's sensor time-series that is changing over time, we can sample positive and negative examples directly from the larger time-series, rather than attempting to create positive examples from augmentations. Unfortunately, the datasets that support this type of task are limited: TNC only included HAR, ECG, and a small simulated dataset. Our work adds the PPG dataset for stress classification, to introduce a new benchmark dataset for this type of time-series contrastive learning task.

Additionally, many of the datasets within UEA and UCR may not suitable for evaluating self-supervised learning, due to their small sizes. Self-supervised learning has risen into prominence due to how it exploits large-scale unlabeled datasets, and such methods have been found to be sensitive to dataset size (Goyal et al., 2019). However, 90 out of the 128 datasets in UCR and 123 out of the 183 datasets in UEA have 1,000 or fewer sequences to train a model. 8 out of the 128 total datasets in UCR and 14 out of the total 183 total datasets in UEA have 100 or fewer sequences. After segmenting our time-series into the contiguous subsequences that are used to train each of our self-supervised learning models, we have a total number of 76,590 / 6,914 / 1,305 sequences for ECG, HAR, and PPG, respectively.

### A.3 BENCHMARK IMPLEMENTATIONS

Our code has been made publicly available at `https://github.com/maxxu05/rebar`. We use the validation set to choose the best model checkpoint to be evaluated on the test set, and tune each model's hyperparameters to achieve its best performance.

For the non-sampling-based methods (i.e. TS2Vec, CPC, SimCLR), we train these methods on sub-sequences. The subsequence sizes used during the contrastive learning stage match the downstream subsequence sizes used for classification: length 128 for HAR, length 3840 for PPG, and length 2500 for ECG. The encoder is kept constant across each of our methods, and we used the encoder from TS2vec (Yue et al., 2022) with an embedding size of 320.

- Fully-Supervised uses the same encoder and linear classification head, trained with class-balanced weights and cross entropy loss, identical to the other linear probe models, only trained end-to-end
    - The PPG and HAR domains were trained with a learning rate of .00001 until convergence, and the ECG domain was trained with a learning rate of .000001 until convergence.
- TS2Vec (Yue et al., 2022) is a current state-of-the-art time-series contrastive learning method that learns time-stamp level representations with augmentations.
    - We follow the default implementation used in `https://github.com/yuezhihan/ts2vec`. In PPG, we have a learning rate of .0001 and batch size of 16, in ECG, we have a learning rate of 0.00001 and batch size of 64, and in HAR, we have a learning rate of 0.00001 and batch size of 64.
- TNC (Tonekaboni et al., 2021) samples a nearby subsequence as a positive and utilizes a hyper-parameter to estimate whether a faraway sampled subsequence is a negative.
    - We follow the default implementation used in `https://github.com/sanatonek/TNC_representation_learning` and set w to .2. in PPG, we have a learning rate of .0001 and batch size of 16, in ECG, we have a learning rate of .0001 and batch size of 16, and in HAR, we have a learning rate of .00001 and a batch size of 16. At the end of the encoder, we utilize a global max pooling layer to pool over time.
    - Note that our reported TNC results are higher than the original reported results because we use TS2Vec's proposed dilated convolution encoder instead of TNC's originally proposed encoder. This was done in order to fairly compare all of the contrastive learning baselines: this encoder backbone constant across all of the baselines. TSVec also found that training TNC with the dilated convolution encoder achieves better results (See Appendix C.3 in (Yue et al., 2022)). Additionally, other discrepancies may originate from how TNC evaluated their method by training a classification head with the encoder model end-to-end, rather than freezing the encoding. We opt to freeze the encoder in order to allow for the encoded representation to be directly evaluated via a linear classifier (i.e. Acc, AUROC, AUPRC) and clusterability (i.e. Adjusted Rand Index and Normalized Mutual Information).

- CPC (Oord et al., 2018) contrasts based on future timepoint predictions.
  - We follow the default implementation used in `https://github.com/jefflai108/Contrastive-Predictive-Coding-PyTorch`. For PPG, we have a learning rate of .001 and batch size of 16, for ECG, we have a learning rate of .0001 and batch size of 16, and for HAR, we have a learning rate of .001 and a batch size of 64.
- SimCLR (Chen et al., 2020) is a simple augmentation-based method we have adapted for time-series. The most common time-series augmentations are used (i.e. shifting, scaling, jittering), allowing us to assess how suitable a pure augmentation-based strategy is.
  - We have 3 augmentations: scaling, shifting, and jittering, with each of the three having a 50% probability of being used. Scaling multiplies the entire time-series with a number drawn from $U(0.5, 1.5)$. Shifting shifts the time-series by a random number between -subsequence_size to subsequence_size. Jittering adds random gaussian noise to the signal, with the Gaussian noise's standard deviation set to .2 of the standard deviation of the values in the entire dataset. For PPG, we have a learning rate of .001, $\tau$ is 1, and batch size of 16, for ECG, we have a learning rate of .001, $\tau$ is .001, and batch size of 16, and for HAR, we have a learning rate of .001, $\tau$ is 1, and batch size of 64. At the end of the encoder, we utilize a global max pooling layer to pool over time.
- Sliding-MSE is a simpler version of REBAR, in which a simple method is used to assess the similarity of a pair of sequences. We slide the candidate subsequence, padded with the true sequence values, along the anchor subsequence and calculate MSE at each iteration. The lowest MSE of the slide is then used as our measure to label the positive and negative examples.
  - For PPG, we have a learning rate of .001, 20 sampled candidates, and $\tau$ is 1000, for ECG, we have a learning rate of .001, 20 sampled candidates, and $\tau$ is .01, and for HAR, we have a learning rate of .001, 20 sampled candidates, and $\tau$ is 0.1. We match the $\alpha$ value used for each dataset, according to the best used by REBAR. At the end of the encoder, we utilize a global max pooling layer to pool over time.
- REBAR is our Retrieval-Based Reconstruction method for time-series contrastive learning
  - Our cross-attention model was trained to convergence and the dilated convolution block has an embedding size of 256 channels, initial kernel size of 15 and dilation of 1. The dilation gets doubled for each following layer. The input channel size of the first convolution is equal to the number of channels present in the data, and the bottleneck layer has size 32. For the PPG and ECG datasets, we have 6 dilated convolution layers to capture a larger receptive field of 883 and for the HAR dataset, we have 2 layers to capture a receptive field of 43. During training, our extended mask sizes for PPG, ECG, and HAR are 300, 300, and 15 respectively. During evaluation, the transient mask masked out 50% of the time points of the signal. During contrastive learning, for PPG data, we have a learning rate of .0001, 20 sampled candidates, $\tau$ is 10, $\alpha$ is 0.5, and batch size of 16, for ECG, we have a learning rate of .001, 20 sampled candidates, $\tau$ is .01, $\alpha$ is 1, and a batch size of 16, and for HAR, we have a learning rate of .001, 20 sampled candidates, $\tau$ is .1, $\alpha$ is 0, and a batch size of 64. At the end of the encoder, we utilize a global max pooling layer to pool over time.

## A.4 EXTRA RESULTS AND DETAILS

### A.4.1 REBAR NEAREST NEIGHBOR VALIDATION EXPERIMENT ALGORITHM

Algorithm 1 is the procedure for generating the confusion matrix from the REBAR Nearest Neighbor Validation Experiment detailed in Sec. 4.3 with Eq. 9. This algorithm uses the notation that has been established throughout the paper, please see Sec. 3 for notation details.

---

**Algorithm 1** Confusion Matrix from REBAR Nearest Neighbor Validation Experiment

---

**Require:** $\mathbf{A} \in \mathbb{R}^{N \times U \times D}$ $\triangleright$ $N$ time-series with length $U$ and dimensionality $D$
**Require:** $C \geq 2$ $\triangleright$ total number of classes, $C$
**Require:** Trials $\geq 1$ $\triangleright$ total number of trials, Trials
**Require:** $0 < T < U$ $\triangleright$ subsequence length, $T$
  **procedure** NEARESTNEIGHBORCONFUSIONMAT($\mathbf{A}$, $C$, Trials, $T$)
    $\mathbf{M} \leftarrow \text{zeros}(C, C)$
    **for** $i$ in range(N) **do**
      **for** $c_{\text{true}}$ in range(C) **do**
        **for** _ in range(Trials) **do**
          # Randomly segment an anchor subseq with $c_{\text{true}}$
          $\mathbf{X}^{(i)}_{\text{anchor},c_{\text{true}}} \leftarrow \text{RandSegment}(\mathbf{A}^{(i)}, T, c_{\text{true}})$
          # Randomly segment a candidate subseq from each class
          $\mathbf{X}^{(i)}_{\text{cand},1} \leftarrow \text{RandSegment}(\mathbf{A}^{(i)}, T, 1)$
          $\vdots$
          $\mathbf{X}^{(i)}_{\text{cand},C} \leftarrow \text{RandSegment}(\mathbf{A}^{(i)}, T, C)$
          **Assert:** $\mathbf{X}^{(i)}_{\text{anchor},c_{\text{true}}}, \mathbf{X}^{(i)}_{\text{cand},1}, \cdots, \mathbf{X}^{(i)}_{\text{cand},C} \in \mathbb{R}^{T \times D}$
          # Nearest neighbor prediction of the anchor's class
          # with distance measured by REBAR measure
          $c_{\text{pred}} \leftarrow \underset{c \in \{1, \cdots, C\}}{\text{argmin}} \, d(\mathbf{X}^{(i)}_{\text{anchor},c_{\text{true}}}, \mathbf{X}^{(i)}_{\text{cand},c})$
          # Update confusion matrix
          $\mathbf{M}[c_{\text{true}}, c_{\text{pred}}] \leftarrow \mathbf{M}[c_{\text{true}}, c_{\text{pred}}] + 1$
        $\mathbf{M}[c_{\text{true}}, :] \leftarrow \mathbf{M}[c_{\text{true}}, :]/\text{Trials}$
    **return M**

---

### A.4.2 FULL t-SNE VISUALIZATIONS

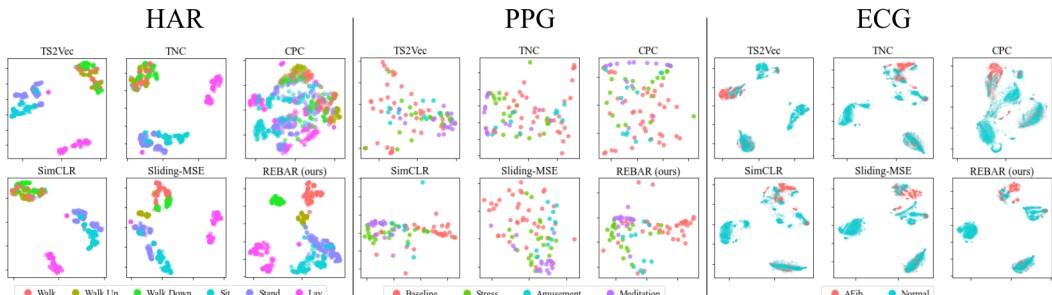

Figure 16: Qualitative Clusterability Results with t-SNE visualizations for all three datasets: HAR, PPG, and ECG. n HAR, all methods except for REBAR group walk, walk up, and walk down together. In PPG, all of the methods have poor clustering, but REBAR seems to perform the best, with only the Meditation label being not clearly disjoint. In ECG, REBAR and TS2Vec qualitatively have the best intra-class clustering.

### A.4.3 INVESTIGATING FALSE POSITIVES AND FALSE NEGATIVES

**False Positives**: Unlike the experiments detailed Sec. 4.3, Fig. 5, and Appendix A.4.1, where 1 subsequence is drawn from each class to be compared to the anchor, in this section, we segment 20 subsequences at randomly from the time-series, which matches our candidate set size used in our contrastive learning experiments for REBAR across all of our datasets, giving us the confusion matrices seen below in Fig. 17. As a result, these 20 candidates can be any distribution of classes.

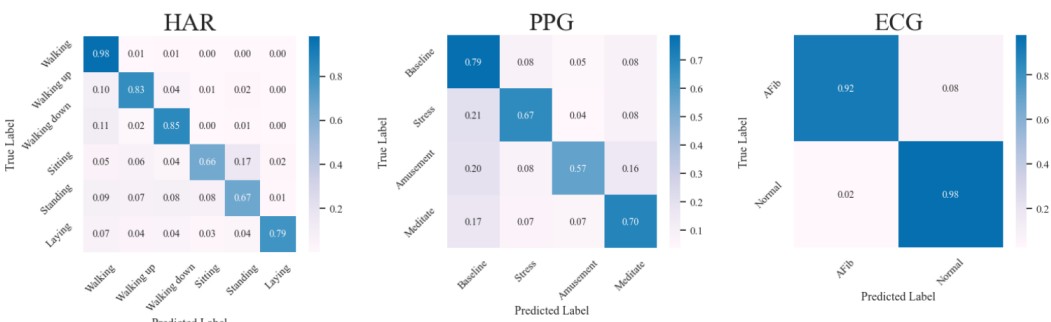

Figure 17: Confusion matrix showing predictions of positive instance being the same class as the anchor given a candidate set size of 20.

We see that with a candidate set size of 20, REBAR achieves a True Positive Rate of predicting the positive instance to be the same class as an anchor of 0.79, 0.68, 0.96 for HAR, PPG, and ECG respectively. The chance of achieving false positives with our candidate size of 20 within our contrastive learning procedure is quite low, and any false positives that do occur are likely to be semantically related to the anchor's true class.

**False Negatives**: False negatives may occur in both the within-ts negatives and/or the between-ts negatives due to the random sampling of candidates and randomness present in the batch, respectively. However, we note that in general, false negatives are a fundamental issue within contrastive learning more broadly and not specific to our REBAR method. For example, in augmentation-based approaches like SimCLR (Chen et al., 2020), negative instances are sampled from the batch, and the batch may contain instances from the same downstream class label as the anchor. However, they are able to still achieve strong performance. Similarly, our approach is still able to achieve the best representation learning performance across our baselines even with the potential presence of false negatives. One idea to address this could be a thresholding method for our within-time-series negatives, but we believe that this method would be ineffective. A threshold should not be designed to be a hyperparameter as it would be dependent on how each specific anchor's specific motifs compare to other general instances within the same class. However, it is not obvious how to construct the threshold as a function of the anchor because during contrastive learning, we do not have access to the ground truth class labels.

### A.4.4 EXTRA VISUALIZATIONS FOR POSITIVE PAIRS

The visualizations from the main text in Fig. 7 and below were randomly selected and each represent a *distinct* random sampling of candidate and anchor subsequences from a different subject's time-series. The accuracy of our positive candidate identification via REBAR is quite high, and each of the figure captions helps describe and justify any misclassifications.

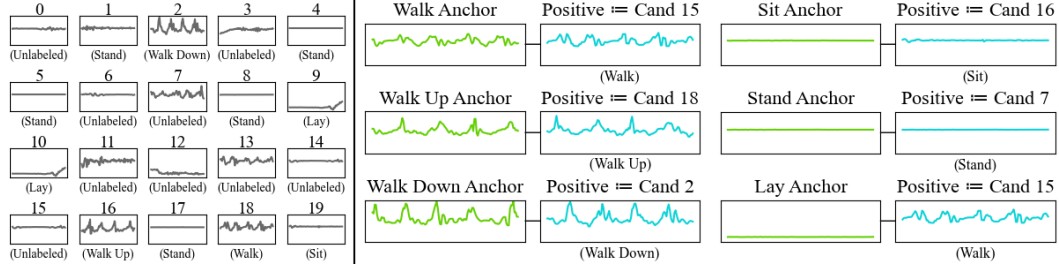

Figure 18: 6/6 anchors of each class are correctly matched with a candidate that shares that class

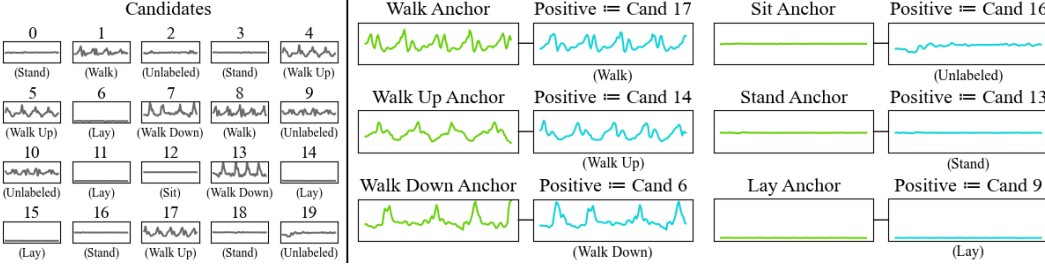

Figure 19: 5/6 anchors are correctly matched. Although candidate 12 is a "Sit" among the 20 candidates, REBAR matches the "Sit" anchor with an "Unlabeled" candidate 16. This seems to be because, even though Candidate 16 is "Unlabeled" it is qualitatively very similar to the anchor.

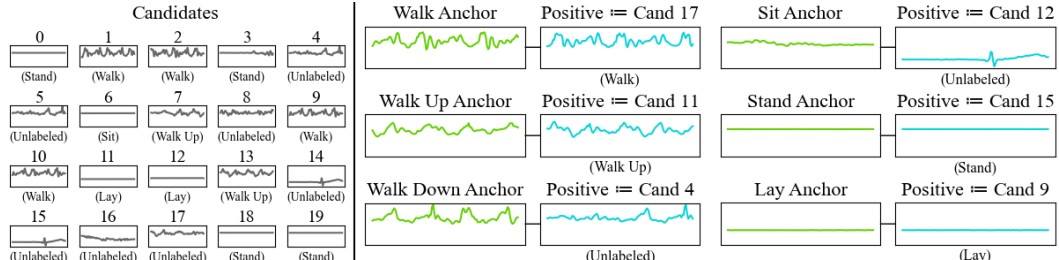

Figure 20: 4/6 anchors are correctly matched. There is no "Walk Down" subsequence among the 20 candidates, but REBAR matches the "Walk Down" anchor with an "Unlabeled" candidate, which still is qualitatively similar to the anchor. The "Sit" anchor is matched with a different "Unlabeled" candidate, which is not ideal. However, there is only 1 "Sit" subsequences among the candidates and matching with an "Unlabeled" class is better than mismatching with a specific different class (e.g. matching "Sit" with "Walk").

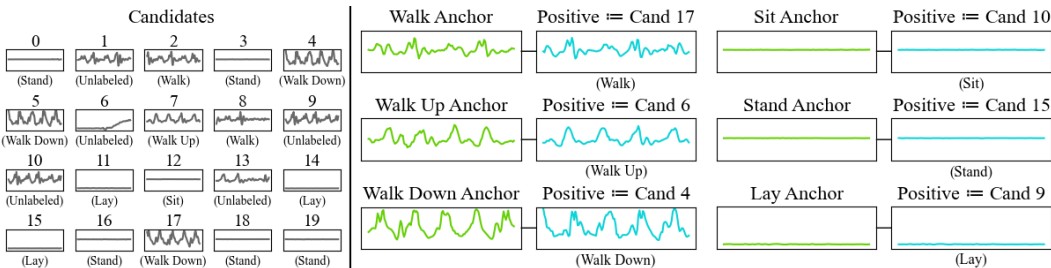

Figure 21: 6/6 anchors of each class are correctly matched with a candidate that shares that class

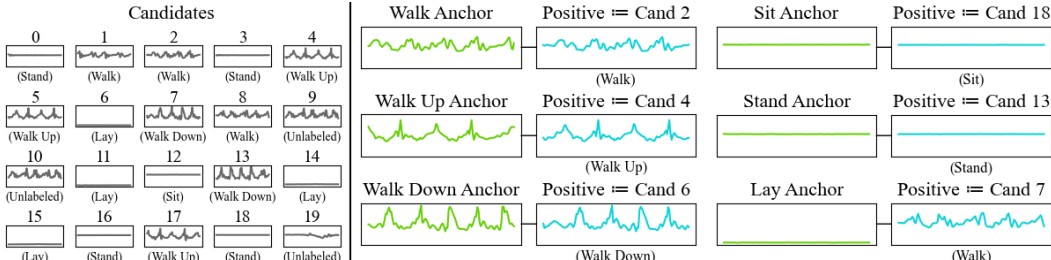

Figure 22: 5/6 anchors are correctly matched. The "Lay" anchor is incorrectly matched with a "Walk" candidate. This may be because it is difficult to do effective motif comparison and retrieval with the flat motifs found in "Lay" query. This is also seen in our Nearest Neighbor experiments shown in Fig. 5, where the flatter signals (i.e. "Sit", "Stand", "Lay") have worse mutual class prediction accuracy compared to the more dynamic signals (i.e. "Walk", "Walk Up", "Walk Down")

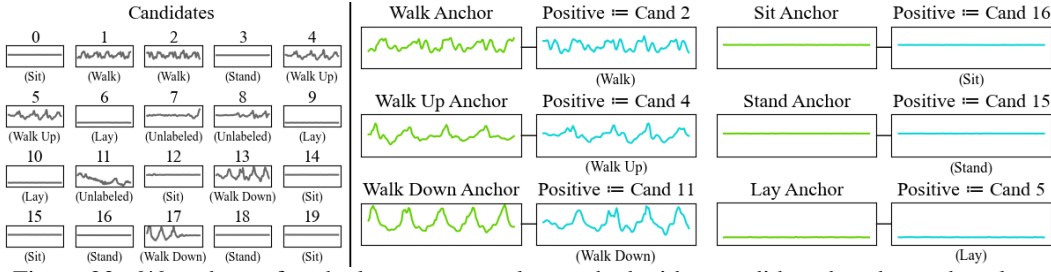

Figure 23: 6/6 anchors of each class are correctly matched with a candidate that shares that class

