# RETRIEVAL-BASED RECONSTRUCTION FOR TIME-SERIES CONTRASTIVE LEARNING

## ABSTRACT

The success of self-supervised contrastive learning hinges on identifying positive data pairs that, when pushed together in embedding space, encode useful information for subsequent downstream tasks. However, in time-series, this is challenging because creating positive pairs via augmentations may break the original semantic meaning. We hypothesize that if we can retrieve information from one subsequence to successfully reconstruct another subsequence, then they should form a positive pair. Harnessing this intuition, we introduce our novel approach: REtrieval-BAsed Reconstruction (REBAR) contrastive learning. First, we utilize a convolutional cross-attention architecture to calculate the REBAR error between two different time-series. Then, through validation experiments, we show that REBAR error is a predictor for mutual class membership, justifying its usage as a positive/negative labeler. Finally, once integrated into a contrastive learning framework, our REBAR method is able to learn an embedding that achieves state-of-the-art performance on downstream tasks across diverse modalities.

## 1 INTRODUCTION

Self-supervised learning uses the underlying structure within a dataset to learn rich and generalizable representations without labels, enabling fine-tuning on various downstream tasks. This reduces the need for large labeled datasets, which makes it an attractive approach for the time-series domain. With the advancement of sensor technologies, it is increasingly feasible to capture a large volume of data, but the cost of data labeling remains high. For example, in mobile health, acquiring labels requires burdensome real-time annotation (Rehg et al., 2017). Additionally, in medical applications such as ECG analysis, annotation is costly as it requires specialized medical expertise.

Contrastive learning is a popular self-supervised learning technique, which involves constructing and embedding positive and negative pairs to yield an embedding space that captures semantic relationships in the data. To be successful, the process for generating such pairs should capture important structural properties of the data. In the vision applications that have driven this approach, augmentations are used to construct a positive pair by exploiting well-established invariances of the imaging process (e.g. flipping and rotating). Unfortunately, general time-series signals do not possess a large and rich set of such invariances. Shifting, which addresses translation invariance, is widely-used, but other augmentations such as shuffling, scaling, or filtering can destroy the signal semantics. For example, shuffling an ECG waveform destroys the temporal structure of the QRS complex, and scaling it can change the clinical diagnosis. Moreover, there is no consensus in the literature, with methods such as TF-C (Zhang et al., 2022) incorporating jittering and scaling augmentations, while TS2Vec (Yue et al., 2022) finds that these augmentations impair performance on downstream tasks.

In this work we introduce a novel approach to identifying positive and negative pairs for contrastive time series learning. In our framework, we define a time-series as being a composition of a series of subsequences, each of which has a class label. This conceptualization describes many real-world biophysical signals. For example, a participant wearing an accelerometer generates a time-series over the course of a day which consists of a series of subsequences of different activities such as sitting and walking. Each subsequence is an instantiation of a specific class of activities, and subsequences with the same class label are likely to have similar temporal patterns. Therefore, two subsequences with similar temporal patterns would be a good candidate to form a positive pair.

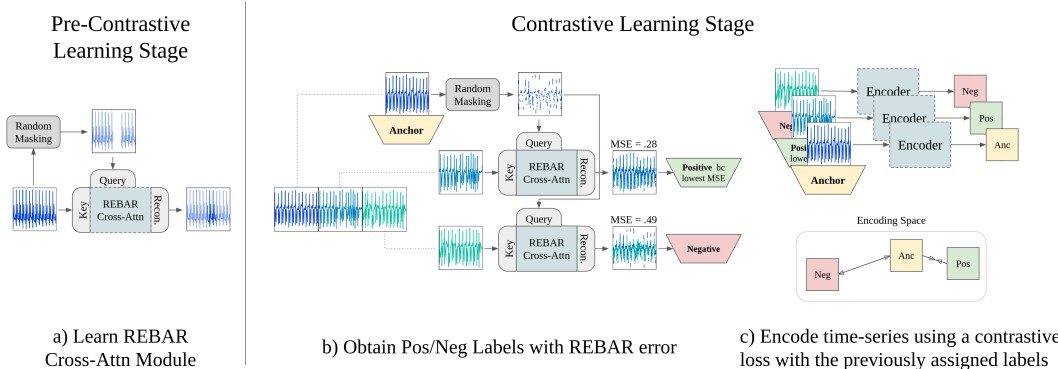

Figure 1: Our REtrieval-BAsed Reconstruction (REBAR) contrastive learning method uses the intuition that if one time-series is useful for reconstructing another, then they likely share reconstruction information and thus should be drawn together within the embedding space as positive examples. a) First, our REBAR cross-attention is trained to retrieve information from the key to reconstruct a masked-out query. b) Next, it is frozen and utilized to identify positive/negative labels. After sampling subsequences from the time-series, the subsequence that reconstructed the anchor with the lowest REBAR error is labeled as positive, and the others are labeled as negative. c) After encoding each of these subsequences, we use the assigned labels within a contrastive loss function.

We propose to measure the similarity between two subsequences by using one subsequence to reconstruct the other one, and taking the reconstruction error as a similarity measure. Pairs with low reconstruction error can then form positive examples in contrastive learning. Because the pair is likely to share the same class label, the resulting learned embedding space is likely to be class-discriminative. We investigate this hypothesis with our REtrieval-BAsed Reconstruction (REBAR) method. In REBAR, a convolutional cross-attention module retrieves information from one subsequence to reconstruct another. We show that the ensuing reconstruction error is a good predictor for mutual class membership, allowing us to identify positive and negative pairs to train our contrastive learning model. This learned embedding space achieves SOTA results with a linear probe and in clusterability. Note that we will interchangeably use "REBAR" to refer to the overall REBAR contrastive learning approach, the REBAR cross-attention module, and the REBAR metric, with the specific meaning being evident from the context. Our major contributions in this work are:

1. We introduce a novel method of identifying positives and negatives for time-series contrastive learning with our REBAR metric, calculated from a convolutional cross-attention architecture that does a between-sequence retrieval and reconstruction of class-specific temporal shapes.
2. To justify this approach and as a prerequisite to contrastive learning, we demonstrate that the REBAR metric is a good predictor for mutual class membership between the two time-series.
3. We benchmark against a representative suite of time-series contrastive learning methods: TS2Vec, one of the current state-of-the-art contrastive learning methods; TNC, which contrasts based on sampled subsequences; CPC, which contrasts based on future predictions; SimCLR, which contrasts based on the most common augmentations; sliding-MSE, a simplified version of our approach. We show that our REBAR contrastive learning approach achieves state-of-the-art performance against all of these baselines with a linear probe evaluation and a cluster agreement check on three distinct domains: ECG, PPG, and HAR.

## 2 RELATED WORKS

**Augmentation-based Contrastive Learning in Time-Series Research:**

Augmentation-based methods are the most widely-studied type of contrastive learning method in time-series research, due to the success of augmentation-based strategies in the computer vision (He et al., 2020; Chen et al., 2020; Chen & He, 2021; Caron et al., 2021). However, it is unclear which augmentation strategies are most effective for time-series, and the findings across different works are inconsistent. For example, TS2Vec (Yue et al., 2022) uses cropping and masking to create positive examples, and their ablation study found that utilizing additional jittering, scaling, and shuffling augmentations led to performance drops. Conversely, TF-C (Zhang et al., 2022) included jittering

and scaling, along with cropping, time-shifting, and other frequency-based augmentations. Similarly, TS-TCC (Eldele et al., 2023) augments the original time-series with either jittering+scaling or jittering+shuffling. This is in spite of the fact that shuffling breaks temporal dependencies and scaling can change the semantic meaning of a bounded signal. Other augmentation works (Woo et al., 2022; Yang & Hong, 2022; Yang et al., 2022b; Ozyurt et al., 2022; Lee et al., 2022) also use some specific combination of scaling, shifting, jittering, and masking. In each case, empirical performance was used to validate the augmentation choice, but differences in datasets, architectures, and training regimes make it difficult to draw a clear conclusion. In contrast, our REBAR method utilizes an alternative approach to generating training samples which is not based on augmentation.

**Sampling-based Contrastive Learning in Time-Series Research:**

A common paradigm in working with time-series data collected over a long period of time (e.g. hours or days) is to partition it into shorter subsequences (e.g. at the level of seconds or minutes, respectively) for labeling and classification. Throughout this paper, we shall refer to the partitions of a longer time-series signal as subsequences. Sampling-based methods attempt to capture the evolving dynamics present in the time-series by sampling subsequences to construct positive pairs.

In Franceschi et al. (2019), an anchor subsequence is sampled from the time-series, and the positive subsequence is cropped from the anchor with the negative cropped from a different time-series. CLOCS (Kiyasseh et al., 2021) samples pairs of temporally-adjacent subsequences and pairs of subsequences across channels from the same time-series as positive pairs. After TNC (Tonekaboni et al., 2021) samples the anchor from the time-series, the positive and unlabeled examples are sampled from neighboring and non-neighboring regions, respectively. The neighboring region is determined via a stationarity test, so the test is ran for every positive/unlabeled sampling, resulting in TNC's slow run-time (250x slower than TS2Vec (Yue et al., 2022)). In addition to this, TNC utilizes a hyperparameter to estimate the probability the unlabeled example is a true negative and does not precisely characterize how stationarity is useful for selecting positive examples. Our REBAR approach is a sampling-based method, but unlike prior works, our positive examples are not simply selected based on temporal proximity to the anchor. Instead, positive examples are selected based on their similarity with the anchor, measured by retrieval-based reconstruction.

**Other Self-supervised Learning Methods in Time-Series Research:**

CPC is a contrastive learning method that trains a network to predict future data points and then learns to contrast them against incorrect ones (Oord et al., 2018). There have also been contrastive learning methods designed for specific time-series modalities that use expert knowledge, such as for EEG (Zhang et al., 2021) or ECG data (Aydemir, 2023).

Another recent self-supervised learning architecture is the MAE, which is used extensively in both NLP (Devlin et al., 2018) and Vision (He et al., 2022), and a few works have adapted it to the time-series domain (Cheng et al., 2023; Li et al., 2023). These models typically utilize transformers to encode the masked input into an embedding space before using a lightweight decoder for reconstruction. While both our REBAR approach and MAEs involve masked reconstruction tasks, their fundamental approaches differ significantly. Given a time-series, our method samples a pair of subsequences and estimates if they are of the same state through a paired reconstruction error, and then, encodes each subsequence based on this estimate. In comparison, MAEs learn to encode a subsequence by understanding how the composition of unmasked time points within the subsequence can be used for reconstruction, and thus, they are not designed to explicitly compare subsequences.

**Retrieval-based Cross-Attention Methods:**

Information retrieval is the process of identifying items from a larger pool of data that are relevant to a given query, often using similarity measures. Our retrieval-based reconstruction model utilizes a cross-attention mechanism to retrieve information across different subsequences for comparison, and there are other related works that utilize cross-attention retrieval. The most common usage of it is in a cross-modality context, in which text is given to the query to retrieve relevant regions in an image (Lee et al., 2018; Miech et al., 2021; Cao et al., 2020; Zheng et al., 2022). Li et al. (2020); Hao et al. (2017) use cross-attention between sentences and words to assess semantic similarity, and Yang et al. (2020) uses a frozen cross-attention network to provide augmentations while training a neural retrieval model in a Q&A task. In the time-series domain, (Garg & Candan, 2021) learns

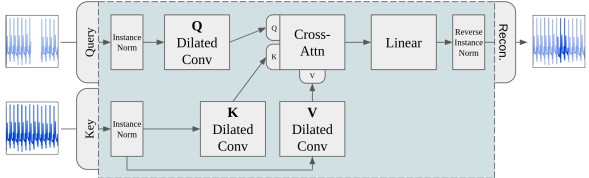

Figure 2: Our REBAR cross-attention module reconstructs the masked query input from the key input. Cross-attention learns a weighted average of a transformation of the key, with the weights as a learned similarity between the query and keys. The attention weights effectively act as a retrieval function, identifying the regions of the key most informative for reconstruction of the query. As such, the query is not directly used for reconstruction: it is only used for retrieval of the keys.

a cross-attention network to retrieve the most salient information from a signal, and (Yang et al., 2022a) utilizes cross-attention to retrieve time-series from a database to assist in forecasting.

While there are self-supervised works that use retrieval tasks to evaluate their learned representation (Chen et al., 2021; Zhu et al., 2020; Ma et al., 2022; Wang et al., 2022), these are very different from our REBAR approach, in which we are directly using a retrieval-based method in an unsupervised fashion to train our self-supervised contrastive learning.

## 3 METHOD

Our approach utilizes retrieval-based reconstruction, REBAR, as a metric for contrastive learning, as it is a strong predictor for mutual class membership between subsequences. The full REBAR contrastive learning approach is shown in Fig 1. Sec 3.1 describes our proposed cross-attention module for learning the REBAR metric, Sec 3.2 explains how we train this model and use it to compare different time-series, Sec 3.3 then investigates the validity of our assumption that REBAR is a good metric for contrastive learning, and Sec 3.4 explains the REBAR contrastive learning.

### 3.1 DESIGN AND IMPLEMENTATION OF THE REBAR CROSS-ATTENTION MODULE

Retrieval-based reconstruction works by retrieving the most useful information in one subsequence to reconstruct another. Cross-attention is an attractive method for modeling this paradigm as the attention weights can be modeled as a retrieval function. Cross-attention learns a weighted average of a transformation of the key time-series, and the weights are a learned similarity between the query and keys. We describe it in our reconstruction setting with its expected value formulation below:

$$\text{CrossAttn}(\bar{\mathbf{x}}_q; \mathbf{X}_k) = \sum_{\mathbf{x}_k \in \mathbf{X}_k} \frac{\exp(\langle f_q(\bar{\mathbf{x}}_q), f_k(\mathbf{x}_k)\rangle)}{\sum_{\mathbf{x}'_k \in \mathbf{X}'_k} \exp(\langle f_q(\bar{\mathbf{x}}_q), f_k(\mathbf{x}_{k'})\rangle)} f_v(\mathbf{x}_k)$$

$$= \sum_{\mathbf{x}_k \in \mathbf{X}_k} p(\mathbf{x}_k | \bar{\mathbf{x}}_q) f_v(\mathbf{x}_k)$$

$$= \mathbb{E}_{p(\mathbf{x}_k | \bar{\mathbf{x}}_q)}[f_v(\mathbf{x}_k)]$$

where $f_q(\bar{\mathbf{x}}_q) = \bar{\mathbf{x}}_q \mathbf{W}_q$, $f_k(\mathbf{x}_k) = \mathbf{x}_k \mathbf{W}_k$, $f_v(\mathbf{x}_k) = \mathbf{x}_k \mathbf{W}_v$, and $\mathbf{W}_{\{q/k/v\}} \in \mathbb{R}^{D_x \times D}$ in the vanilla model. $\mathbf{X}_k \in \mathbb{R}^{T \times D_x}$ is the key subsequence. $\bar{\mathbf{x}}_q^\top \in \mathbb{R}^{D_x}$ is a row in $\bar{\mathbf{X}}_q \in \mathbb{R}^{T \times D_x}$, which is a masked version of $\mathbf{X}_q$ query subsequence. We omit the scaling factor and biases for brevity.

Given a query, a retrieval function searches for the most useful information within a potentially different database that can be used for some downstream task. In our setting, this query is a query subsequence, the database is a different key subsequence, and the downstream task is reconstruction. With a cross-attention model, $p(\mathbf{x}_k | \bar{\mathbf{x}}_q)$ is the data retrieval function and the transformation of the keys, $f_v(\mathbf{x}_k)$, allows us to reconstruct from the retrieved data. However, if we use the raw subsequence, vanilla $f_{k/q}$ is calculated with only a single time-point of information, and thus, our retrieval function, $p(\mathbf{x}_k | \bar{\mathbf{x}}_q)$, can only compare single time points with each other. A single time step holds limited information about its semantic content, so, following Xu et al. (2022), we utilize a dilated convolution $f_{k/q/v}$ to captures features from the wider surrounding temporal neighborhood

to calculate each $p(\mathbf{x}_k|\bar{\mathbf{x}}_q)$ and to enable retrieval and reconstruction of larger temporal shapes. Our full REBAR$(\bar{\mathbf{X}}_q, \mathbf{X}_k)$ cross-attention module is visualized in Fig 2.

We opt to not use any positional encoding because we would like the retrieval function, $p(\mathbf{x}_k|\bar{\mathbf{x}}_q)$, to retrieve regions in the key subsequence based on similarity of the temporal shapes within the key and query, rather than considering the specific positions of such structures. Knowing the exact positions is uninformative due to the arbitrary start-time at which sensors begin recording and are segmented, and thus we seek to instead model temporal relationships through the dilated convolution that can capture the shape of a subsequence. In order to handle the masks, we utilize partial convolutions that ignore the calculation at masked time points (Liu et al., 2018). The linear layer following the cross-attention module is used to project the subsequence from the embedding space back to its original dimension. We use an instance norm to normalize the query and keys, normalized to the query, then reverse it at the end for reconstruction (Kim et al., 2021).

We intentionally utilize a simple approach in order to not obfuscate the query reconstruction. This allows for the reconstruction to be based on using a simple transformation of the raw key subsequence, allowing for the emphasis to be on learning a good retrieval function, $p(\mathbf{x}_k|\bar{\mathbf{x}}_q)$. This retrieval function is the only place in which the query subsequence is used. With an abuse of notation:

$$\text{REBAR}(\bar{\mathbf{X}}_q, \mathbf{X}_k) \perp \bar{\mathbf{X}}_q \mid p(\mathbf{x}_k|\bar{\mathbf{x}}_q)$$

Therefore, it is important to note that *the query subsequence is not directly used for this reconstruction, it is only used to identify regions in the key subsequence to retrieve with $p(\boldsymbol{x}_k|\bar{\boldsymbol{x}}_q)$.* By using cross-attention to model our query reconstruction, we cannot "cheat" by borrowing information from the query itself for reconstruction. The reconstruction information only flows from the retrieved regions of the key subsequence with $f_v(\mathbf{x}_k)$.

## 3.2 Training and Applying REBAR Cross-Attention

Our REBAR approach's intuition says that if we can successfully reconstruct $\bar{\mathbf{X}}_q$ with retrieved information from $\mathbf{X}_k$, then we hypothesize that they likely share reconstruction information and downstream classification labels. Therefore, if $\|\text{REBAR}(\bar{\mathbf{X}}_q, \mathbf{X}_k) - \mathbf{X}_q\|_2^2$ is small, then it predicts whether $y(\mathbf{X}_q) = y(\mathbf{X}_k)$, where $y(\mathbf{X})$ is the true class label of $\mathbf{X}$. If this holds true, then we can utilize REBAR error to identify positive/negative pairs and learn a class discriminative space.

**In the pre-contrastive learning stage,** we must first train our REBAR$(\bar{\mathbf{X}}_q, \mathbf{X}_k)$ to learn how to effectively retrieve and reconstruct class-discriminative information across subsequences. Unfortunately, we do not have $y(\cdot)$, so given a random $\bar{\mathbf{X}}_q, \mathbf{X}_k$ pair, it is unclear whether we want to minimize or maximize $\|\text{REBAR}(\bar{\mathbf{X}}_q, \mathbf{X}_k) - \mathbf{X}_q\|_2^2$. Therefore, we

$$\text{Train } \text{REBAR}(\bar{\mathbf{X}}_q, \mathbf{X}_k) \text{ with } \mathbf{X}_q = \mathbf{X}_k$$

because $y(\mathbf{X}_q) = y(\mathbf{X}_k)$ when $\mathbf{X}_q = \mathbf{X}_k$. The key has exactly the information that the query is missing, so we expect for our model to learn how to retrieve the exact matching region in the key.

Now we must identify the optimal masking strategy so that REBAR cross-attention can learn to capture and retrieve class-specific shapes. We define a binary mask as $m \in \mathbb{R}^T$ with missingness as 1 and non-missingness as 0. We then investigate two types of masks: *contiguous* and *intermittent*. A contiguous mask is where a contiguous segment of length $n$ is randomly masked out (i.e. $m[i : i + n] = 1$ with $i \in_R \{0, 1, \cdots, T - n\}$ and $\in_R$ is random sampling without replacement). An intermittent mask is where $n$ time stamps are randomly masked out (i.e. $m[i_0] = 1, m[i_1] = 1, \cdots, m[i_{n-1}] = 1$ for $i_0, i_1, \cdots, i_{n-1} \in_R \{0, 1, \cdots, T - 1\}$)

We would like the attention weights in the REBAR cross-attention to learn to retrieve and reconstruct from shapes that are indicative of downstream class labels. For example, our model that should retrieve an entire heartbeat for ECG signal reconstruction. Reconstructing a normal signal from one with misshapen heartbeats will have a higher error than reconstructing from another normal signal. If the model is only trained to reconstruct transient trends, like a short upwards slope, both normal and abnormal signals will exhibit this trend. This will then lead to similar reconstruction performance, thereby reducing the model's discriminative capability. Thus, we would like to use a contiguous mask during training, which masks out entire temporal shapes.

Fig 3a) shows that the attention weights learned with a contiguous mask are sparse and fit exactly onto the region in the key corresponding to the masked region in the query. If we use an intermittent

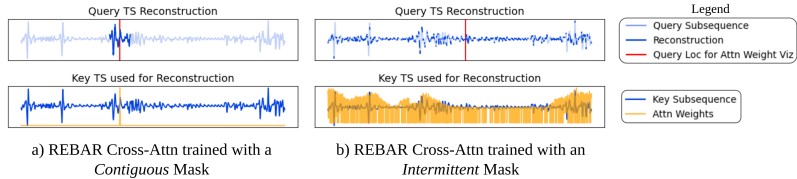

a) REBAR Cross-Attn trained with a
*Contiguous* Mask

b) REBAR Cross-Attn trained with an
*Intermittent* Mask

Figure 3: a) While using an contiguous mask to train reconstruction of the query subsequence, the attention weights learned are sparse and specific to the corresponding region in the key subsequence. Conversely, when learned with a intermittent mask, b), the attention weights fit onto transient trends in the key. Specific attention weights allow for an effective comparison of class-discriminative shapes between differing subsequence, so we utilize a contiguous mask for training REBAR.

mask, the model matches to many different temporal shapes throughout the key, which is shown in Fig 3b)'s dispersed attention distribution. This is an issue because the model is unable to retrieve class-specific temporal shapes, choosing instead to match with minor, transient trends. Utilizing a contiguous mask encourages the model to learn to attend to specific temporal shapes in the key.

**In the contrastive learning stage,** we can use our trained REBAR cross-attention to identify whether an anchor and a candidate subsequence should form a positive or negative pair:

$$\text{Evaluate } \|\text{REBAR}(\bar{\mathbf{X}}_{anchor}, \mathbf{X}_{cand}) - \mathbf{X}_{anchor}\|_2^2 \text{ with } \mathbf{X}_{anchor} \neq \mathbf{X}_{cand}$$

With our trained REBAR cross-attention, the model will attempt to reconstruct $\bar{\mathbf{X}}_{anchor}$ from $\mathbf{X}_{cand}$, and we can use the reconstruction error to predict whether the two subsequences share a mutual class and use this to label a pair as being either positive or negative.

During evaluation, we use an an intermittent mask. Fig 5 shows that the model trained with a contiguous mask still maintains a sparse attention when evaluated on a transient mask. Then, because the masked time points are interspersed throughout the signal in an intermittent mask, we are able to evaluate retrieval-reconstruction at many different parts of the signal for a better comparison between two different subsequences. The combination of using a contiguous mask for training and an intermittent mask for application is validated empirically in Appendix A.1.

## 3.3 UNDERSTANDING AND VALIDATING REBAR

Before integrating the REBAR metric into a contrastive learning framework, we assess whether it effectively predicts mutual class membership by borrowing downstream classification labels for a validation experiment. Our data is structured such that there are $n$ time-series, $\mathbf{A} \in \mathbb{R}^{U \times D}$, where $U$ is length and $D$ is channel. We segment each one into short subsequences, $\mathbf{X} \in \mathbb{R}^{T \times D}$ with $T \ll U$. The time-series $\mathbf{A}$ has changing dynamic states (e.g. a PPG signal that measures fluctuating stress states) over time so that a given subsequence $\mathbf{X}$ has a specific label (e.g. stressed, not stressed).

From the time-series, we randomly segment out one anchor subsequence, $\mathbf{X}_{anchor}$ and $C$ candidate subsequence, $\mathbf{X}_{cand}^0, \cdots, \mathbf{X}_{cand}^C$, from each of the $C$ classes. Then, the anchor subsequence's class is predicted to be the class of the candidate that provided the lowest REBAR error:

$$\text{pred}(\mathbf{X}_{anchor}) = \underset{i}{\arg\min} \|\text{REBAR}(\bar{\mathbf{X}}_{anchor}, \mathbf{X}_{cand}^i) - \mathbf{X}_{anchor}\|_2^2$$

The above procedure is then bootstrapped and repeated for each time-series to obtain an average of the class prediction accuracy. The final results are visualized in the confusion matrices found in Fig 4. The relatively high concentration on the diagonal in our confusion matrices across our different datasets demonstrates that REBAR, although it is trained with a masked reconstruction task without any class labels, is still able to predict mutual class membership. This validates our approach for using REBAR as a metric for contrastive learning.

Fig 5 shows the reconstruction of an anchor subsequence with different candidate subsequences. Fig 5b) and c) show that despite being trained in a setting in which the key and the query are the same instance, the REBAR model is still able to reconstruct the query when the key is different. The attention weights retrieve the most relevant information from the key, and the weights being sparse indicates that specific temporal shapes are being retrieved.

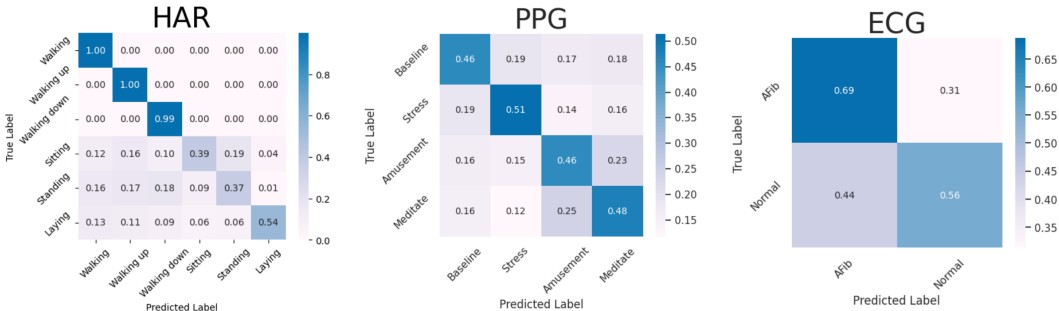

Figure 4: The strong diagonal pattern observed across all three datasets' confusion matrices highlights REBAR's ability to predict mutual class membership. Before its use in contrastive learning, REBAR's effectiveness in identifying pos/neg examples can be evaluated by leveraging downstream labels. Given an anchor subsequence, we sample a candidate subsequence from each class, and the anchor subsequence is predicted to be the class of the candidate with the lowest REBAR error with a masked-out anchor. The mean of the bootstrapped accuracies is shown in the confusion matrices.

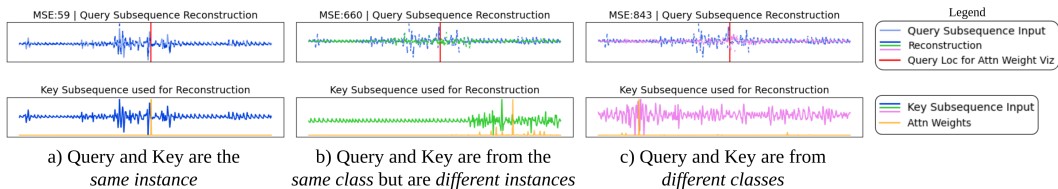

a) Query and Key are the *same instance*

b) Query and Key are from the *same class* but are *different instances*

c) Query and Key are from *different classes*

Figure 5: Visualization of REBAR cross-attention results for three different scenarios. a) The query and key are the same instance, so all of the information masked out in the query can be retrieved from the key. b) and c) utilize keys that differ from the query, but our REBAR cross-attention still attempts to retrieve the most relevant information in the key for reconstruction. The orange attention weights are conditioned on a downwards signal dip in the query, shown by the red vertical line, and each of the attention weights across a) b) c) are highest for a corresponding dip in the key subsequence. Because b)'s key is in the same class as the query, the MSE in b) is lower than c).

## 3.4 APPLYING OUR REBAR METRIC IN CONTRASTIVE LEARNING

With our trained $\text{REBAR}(\bar{\mathbf{X}}_q, \mathbf{X}_k)$, we can apply it within a contrastive learning framework. Given a time-series, we uniformly at random sample an anchor subsequence, $\mathbf{X}_{anchor}$, and $n$ candidate subsequences, $\mathbf{X}_{cand}$, that make up the set $\mathcal{S}_{cand}$. We can then use REBAR to label our candidates:

$$\mathbf{X}_{pos} = \underset{\mathbf{X}_{cand} \in \mathcal{S}_{cand}}{\operatorname{argmin}} \|\text{REBAR}(\bar{\mathbf{X}}_{anchor}, \mathbf{X}_{cand}) - \mathbf{X}_{anchor}\|_2^2$$

$$\mathcal{S}_{neg} = \mathcal{S}_{cand} \setminus \{\mathbf{X}_{pos}\}$$

We can then use these labels with the NT-Xent loss, $\mathcal{L}$, shown below.

$$\mathcal{L} = \frac{\exp(\text{sim}(\mathbf{X}_{anchor}, \mathbf{X}_{pos})/\tau)}{\exp(\text{sim}(\mathbf{X}_{anchor}, \mathbf{X}_{pos})/\tau) + \sum_{\mathbf{X}_{neg} \in \mathcal{S}_{within-neg}} \exp(\text{sim}(\mathbf{X}_{anchor}, \mathbf{X}_{neg})/\tau)}$$

with $\tau$ as the temperature parameter and $\text{sim}(\cdot, \cdot)$ as the cosine similarity function.

This loss learns to pull the anchor and the subsequence that is most likely to be of the same class as the anchor, according to REBAR, together in the embedding space, while pushing the anchor and the subsequences with higher REBAR error apart. This allows us to capture the differences in states between subsequences in our embedding space. Our full REBAR approach is visualized in Fig 1.

## 4 EXPERIMENTS

Our work seeks to understand how different contrastive learning methods learn representations, so we utilize multiple evaluation methods on the learned encoding and each of the baselines were

selected to represent distinct contrastive learning method paradigms. Further elaboration on these aspects is provided in Sec 4.1. We also incorporate diverse time-series datasets from various domains to assess the performance of each method across different contexts, detailed in Sec 4.2. In Sec 4.3, we detail our results, analyzing each of our baselines and describing how our REBAR method is able to achieve SOTA. Finally, in Sec 4.4, we detail avenues for future work.

## 4.1 BENCHMARKS

This work seeks to identify the best approach to time-series contrastive learning, so the encoder is kept as a control across all baselines. We utilize the dilated convolution encoder utilized by TS2Vec (Yue et al., 2022), and for the methods that have set subsequence lengths, we set it to the subsequence length used in the downstream classification. For benchmarks, we include TS2Vec (Yue et al., 2022), TNC (Tonekaboni et al., 2021), CPC (Oord et al., 2018), SimCLR (Chen et al., 2020), Sliding-MSE, and our method REBAR. Each of these methods are chosen to represent specific time-series contrastive learning modeling paradigms:

- TS2Vec is a current state-of-the-art time-series contrastive learning method that learns time-stamp level representations with augmentations.
- TNC is a sampling-based method that utilizes a hyperparameter to estimate whether an unlabeled candidate is a true negative and assumes all nearby subsequences are positive.
- CPC is a contrastive learning method and elects to contrast based on future timepoint predictions.
- SimCLR is a simple augmentation-based method we have adapted for time-series. We use the most common time-series augmentations, shifting, scaling, and jittering, allowing us to assess how suitable a pure augmentation-based strategy is for time-series contrastive learning.
- Sliding-MSE is a simpler version of REBAR, in which a simple method is used to assess the similarity of a pair of sequences. We slide the candidate subsequence, padded with the true sequence values, along the anchor subsequence and calculate MSE at each iteration. The lowest MSE of the slide is then used as our metric to label the positive and negative examples.

We utilize the validation set to choose the best model checkpoint to be evaluated on the test set, and tune each model's hyperparameters to achieve their best performance. For downstream classification, we learn a linear probe (i.e. logistic regression) on each model's frozen encoding. We then also assess cluster agreement. First, we do a $k$-means clustering on each of the encodings, where $k$ is the number of classes, and then we can assess the similarity of these clusters with the true class labels with the Adjusted Rand Index (ARI) and the Normalized Mutual Information (NMI) metrics.

## 4.2 DATASETS

We utilize 3 datasets from 3 different sensor domains with time-series that will have their classification labels change over time: HAR, PPG, and ECG. Each of these signals have drastically different structures and patterns, and the class-specific temporal shapes vary differently within a modality (e.g. in HAR, differences in labels manifest via amplitude differences; in ECG, differences in labels manifest via frequency differences). The datasets are split into a 70/15/15 train/val/test split.

**HAR**: Rather than using the extracted time and frequency features, we opt to use the raw accelerometer and gyroscopic sensor data (Reyes-Ortiz & Parra, 2015) to better assess how our methods perform on raw time-series. Each time-series is sampled at 50 Hz and has 6 channels. There are 59 5-minute-long time-series, and downstream classification is done on non-overlapping 2.56-second-long subsequences, with the following labels: walking (17.7%), walking upstairs (7.6%), walking downstairs (9.1%), sitting (18.2%), standing (20.1%), and laying (20.1%).

**PPG**: From the WESAD dataset (Schmidt et al., 2018), we utilize the single-channel PPG data sampled at 64 Hz, denoised following the process in Heo et al. (2021). In total, there are 15 total patients with 87-minute-long time-series, with downstream classification on non-overlapping 1-minute-long subsequences. The ground truth of the stress labels were determined by the study protocol, and the labels are baseline (42.7%), stress (24.0%), amusement (12.4%), and meditation (20.9%).

**ECG**: From the MIT-BIH Atrial Fibrillation dataset (Moody, 1983), we utilize the dual-channel ECG data sampled at 250 Hz. In total, we have 23 total patients with 9.25-hour-long time-series, with downstream classification on non-overlapping 10-second-long subsequences. The labels are atrial fibrillation (41.7%) and normal (58.3%).

## 4.3 RESULTS

Our REBAR method learns the best representation across all of our benchmarks and settings. Tbl 1 shows that the linear probe trained on our REBAR representation consistently achieved the strongest results. Our REBAR method demonstrates a much stronger performance than Sliding-MSE, showing the necessity of learning a retrieval-reconstruction model. Our improved performance compared to TNC highlights the value of identifying positives that are not necessarily near the anchor subsequence. The poor results from SimCLR in PPG and ECG underscore the non-trivial nature of designing an augmentation-based method, even when such augmentations are commonly employed. We note that although TS2Vec is a SOTA method for contrastive learning, it is unable to consistently achieve strong performance, and we believe that this is because TS2Vec was evaluated on short time-series that had one label and not label-specific subsequences. The other sampling-based methods (e.g. TNC, Sliding-MSE, REBAR) are able to exploit the data structure in our task to borrow subsequences across the time-series to achieve stronger performance.

| Model | PPG | | | HAR | | | ECG | | |
|---|---|---|---|---|---|---|---|---|---|
| | Accuracy | AUROC | AUPRC | Accuracy | AUROC | AUPRC | Accuracy | AUROC | AUPRC |
| TS2Vec | 0.4023 | 0.6428 | 0.3959 | 0.9324 | 0.9931 | 0.9766 | 0.6889 | 0.7251 | 0.6617 |
| TNC | 0.2989 | 0.6253 | 0.3730 | 0.9437 | 0.9937 | 0.9788 | 0.7175 | 0.8429 | 0.7793 |
| CPC | 0.3448 | 0.5843 | 0.3642 | 0.8662 | 0.9867 | 0.9438 | 0.6856 | 0.6855 | 0.6369 |
| SimCLR | 0.3448 | 0.6119 | 0.3608 | 0.9465 | 0.9938 | 0.9763 | 0.6752 | 0.6618 | 0.5982 |
| Sliding-MSE | 0.3333 | 0.6456 | 0.3831 | 0.9352 | 0.9931 | 0.9767 | 0.6858 | 0.7292 | 0.6345 |
| REBAR (ours) | **0.4138** | **0.6977** | **0.4457** | **0.9535** | **0.9965** | **0.9891** | **0.7928** | **0.8462** | **0.8051** |

Table 1: Linear Probe Classification Results

Tbl 2 shows that when measuring the cluster agreement with the true class labels, REBAR contniues to achieves the best ARI and NMI, corroborating the strong classification results. This is unlike other methods, such as TS2vec in PPG, that achieve strong linear probe results, but low cluster agreement.

| Model | PPG | | HAR | | ECG | |
|---|---|---|---|---|---|---|
| | ARI | NMI | ARI | NMI | ARI | NMI |
| TS2Vec | -0.0353 | 0.1582 | 0.4654 | 0.6115 | 0.0251 | 0.0193 |
| TNC | 0.0958 | 0.1666 | 0.4517 | 0.5872 | -0.0302 | 0.1009 |
| CPC | 0.1110 | 0.1867 | 0.1603 | 0.2217 | 0.0239 | 0.0169 |
| SimCLR | 0.1535 | 0.3081 | 0.5805 | 0.6801 | -0.0372 | 0.1157 |
| Sliding-MSE | 0.1083 | 0.2141 | 0.5985 | 0.7019 | -0.0081 | 0.0005 |
| REBAR (ours) | **0.1830** | **0.3422** | **0.6258** | **0.7721** | **0.4194** | **0.2696** |

Table 2: Similarity between $k$-means Encoding Clusters and Class Labels

## 4.4 FUTURE WORK

Our REBAR method demonstrates that retrieval-based reconstruction can be used as a metric for contrastive learning, and more research can be done on further leveraging this metric. Our current implementation utilizes the REBAR metric as a method of binarizing the data into positives and negatives, but future work could explore how to directly use the REBAR metric such that the distance of pairs in embedding space reflects the difference of their REBAR metrics, similar to what was done in the Log-ratio Loss (Kim et al., 2019). Alternatively, we can enforce the relative order of candidates given by REBAR, by using methods such as ROUL (Kan et al., 2021).

## 5 CONCLUSION

Our Retrieval-Based Reconstruction idea is a novel approach to time-series contrastive learning. By using a cross-attention module to retrieve information in subsequence to reconstruct class-specific shapes in another, we can predict mutual class membership. Then, we have demonstrated that if we utilize this REBAR metric to identify positive and negative pairs in a contrastive learning approach, we are able to achieve state-of-the-art results on learning a class-discriminative embedding space. Our REBAR method offers a new perspective on what a positive pairing in time-series entails, and we hope to drive future research into understanding how to best learn representations in this space.

## 6 ETHICS STATEMENT

Our paper works on creating models for health-related signals, and it has the potential to improve health outcomes, but at the same time could lead to a loss of privacy and could possibly increase health-related disparities by allowing providers to characterize patients in more fine-grain ways. In the absence of effective legislation and regulation, patients may have no control over how their data is being used, leading to questions of whether autonomy and justice, key pillars of medical ethics, are being upheld. Overall though, we hope that our work leads to a net positive as it helps further the field towards creating personalized health recommendations, allowing patients to receive improved care and achieve better health outcomes, directly contributing to patient safety and overall well-being.

## 7 REPRODUCIBILITY STATEMENT

Our Methods section in Section 3 details the way in which we set-up our method, and our Experiments section in Section 4 details our experimental design. Additionally, in the Appendix A.2, we itemize each of the hyperparameters we used to tune each of our benchmarks. In the interest of anonymity, we have not yet released our Github code. Upon acceptance, we will release it to the public, which will have the set seeds and exact code we used to run our experiments. We will also make our model checkpoints downloadable. The datasets used are publicly available, and we describe how we curate each of them for our task in Section 4.2, and upon acceptance, we will also release our specific data preprocessing code + direct the data download.

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

# A APPENDIX

## A.1 MASKING VALIDATION

| Training Mask Type
Evaluation Mask Type | Contiguous
Contiguous | Contiguous
Intermittient | Intermittient
Intermittient | Intermittient
Contiguous |
|---|---|---|---|---|
| Accuracy | 0.3624 | **0.4580** | 0.2888 | 0.2378 |

Table 3: Comparison of Different Masking Procedures for REBAR. The accuracy reported is from the validation strategy explicated in Section 3.3, applied to the PPG dataset, averaged across all classes.

## A.2 MODEL IMPLEMENTATIONS

All of our code with the model implementations and evaluations, along with trained checkpoints and the set seed, will be made publicly available for reproducibility. The subsequences sizes used during the contrastive learning stage match the downstream subsequence sizes used for classification: length 128 for HAR, length 3840 for PPG, and length 2500 for ECG. The encoder is kept constant across each of our methods, and we used the encoder from TS2vec(Yue et al., 2022) with an embedding size of 320.

- **REBAR:** Our cross-attention model was trained to convergence and the bottleneck dilated convolution block has an an embedding size of 256 channels, initial kernel size of 15 and dilation of 1 and then the dilation gets doubled for each following layer. The input channel size of the first convolution is equivalent to the number of channels present in the data, and the bottleneck layer has an embedding size of 32. For the PPG and ECG datasets, we have 6 dilated convolution layers to capture a larger receptive field of 883 and for the HAR dataset, we have 2 layers to capture a receptive field of 43. During training, our contiguous mask sizes for PPG, ECG, and HAR are 300, 300, and 15 respectively. During evaluation, the intermittent mask masked out 50% of the time points of the signal. During contrastive learning, for PPG data, we have a learning rate of .0001, 20 sampled candidates, $\tau$ is 10, and batch size of 16, for ECG, we have a learning rate of .1, 20 sampled candidates, $\tau$ is 1, and a batch size of 16, and for HAR, we have a learning rate of .001, 20 sampled candidates, $\tau$ is .1, and a batch size of 64. At the end of the encoder, we utilize a global max pooling layer to pool over time.

- **Sliding-MSE:** For PPG, we have a learning rate of .001, 20 sampled candidates, and $\tau$ is 1000, for ECG, we have a learning rate of .1, 20 sampled candidates, and $\tau$ is 1000, and for HAR, we have a learning rate of .001, 20 sampled candidates, and $\tau$ is 0.1. At the end of the encoder, we utilize a global max pooling layer to pool over time.

- **SimCLR:** We have 3 augmentations: scaling, shifting, and jittering, with each of the three having a 50% probability of being used. Scaling multiples the entire time-series with a number uniformly sampled from 0.5 to 1.5. Shifting will shift the time-series by a random number between -subsequence_size to subsequence_size. Jittering will add a random gaussian noise to the signal, with the gaussian noise's standard deviation set to .2 of the standard deviation of the values in the entire dataset. For PPG, we have a learning rate of .001, $\tau$ is 1, and batch size of 16, for ECG, we have a learning rate of .001, $\tau$ is 1, and batch size of 16, and for HAR, we have a learning rate of .001, $\tau$ is 1, and batch size of 64. At the end of the encoder, we utilize a global max pooling layer to pool over time.

- **CPC:** We follow the default implementation used in `https://github.com/jefflai108/Contrastive-Predictive-Coding-PyTorch`. For PPG, we have a learning rate of .001 and batch size of 16, for ECG, we have a learning rate of .001 and batch size of 16, and for HAR, we have a learning rate of .001 and a batch size of 64.

- **TNC:** We follow the default implementation used in `https://github.com/sanatonek/TNC_representation_learning` and set w to .2. in PPG, we have a learning rate of .0001 and batch size of 16, in ECG, we have a learning rate of .001 and

batch size of 16, and in HAR, we have a learning rate of .00001 and a batch size of 16. At the end of the encoder, we utilize a global max pooling layer to pool over time.

- **TS2Vec:** We follow the default implementation used in `https://github.com/yuezhihan/ts2vec`. In PPG, we have a learning rate of .0001 and batch size of 16, in ECG, we have a learning rate of 0.00001 and batch size of 64, and in HAR, we have a learning rate of 0.00001 and batch size of 64.