# OpenReview forum: "REBAR: Retrieval-Based Reconstruction for Time-series Contrastive Learning"
_ICLR.cc/2024/Conference — ICLR 2024 poster_

### Official Review · Reviewer_Hx4L · 2023-10-20

**Soundness:** 3 good
**Presentation:** 3 good
**Contribution:** 3 good
**Rating:** 8
**Confidence:** 4

**Summary:**

This paper proposes a time-series contrastive learning framework that replaces the data augmentation module with a retrieval-based pair construction strategy. The idea sounds interesting and is proved to be effective on three time-series datasets.

**Strengths:**

1. This work proposes a retrieval-based mask reconstruction strategy to help the model identify similar time series, which I think is a smart design.
2. The authors show that using contiguous and intermittent masks during the training and evaluation respectively leads to the best performance. Such a result could bring some new insights to the time-series learning community.
3. By constructing contrastive pairs retrieval, the proposed method does not rely on data augmentations, which could harm the pattern of signals, to perform contrastive learning. Experiments on three datasets demonstrate the effectiveness of the proposed method.

**Weaknesses:**

1. Figure 4 shows that the diagonal pattern is worse on the PPG and ECG data compared with that on the HAR data. Some explanations need to be provided here to help readers understand the potential limitations of the method.
2. During the contrastive learning stage, the positive counterpart is selected as the one most similar to the anchor. However, it is possible that there is more than one candidate that shares the same class label with the anchor. Would such false negative pairs influence the performance of contrastive learning? Have you tried other positive selecting strategies such as hard threshold?
3. Typo: we uniformly at random sample -> we uniformly random sample

**Questions:**

Please refer to the weaknesses.

---

> ### Author Response · Authors · 2023-11-21
> **Response to Hx4L (1/1)**
>
> Thank you for your kind words and discussion points. We have revised our text to capture our discussion on the different performance across datasets and the presence of false negatives. We have addressed each of your specific points below (with W = weakness and Q = question).
>
> "_Figure 4 shows that the diagonal pattern is worse on the PPG and ECG data compared with that on the HAR data. Some explanations need to be provided here"_ (W1):
>
> - This is a good point. The different classes within HAR are arguably more distinctive than the classes for ECG and PPG, which can be seen in each dataset's visualization in Fig. 10-12, and we added this detail in the Fig. 5 [formerly Fig. 4] caption. Appendix A.3 further details each of the distinctive domains that each dataset is capturing.
>
> False Negatives (Q2):
>
> - The idea that false negatives potentially being among the candidates is a great discussion point, and it is a fundamental issue within contrastive learning more broadly, rather than being specific to our REBAR method. For example, in augmentation-based approaches like SimCLR [1], they sample negative items from the batch, and the batch may contain items from the same downstream classification category as the anchor. However, they are able to still achieve strong performance. Similarly, our approach is still able to achieve the best representation learning performance across our baselines. In regards to the proposed thresholding method, we believe that this method would be ineffective. A threshold should not be designed to be a hyperparameter as it would be dependent on how each specific anchor's specific motifs compare to other general instances within the same class. However, learning a threshold as a function of the anchor would be non-obvious to do due to the lack of class label ground truth during contrastive learning. Addressing false negatives is an ongoing research area, and we are exploring other strategies as a part of future work. We add this discussion into the Appendix A.6.
>
> Typo (Q3):
>
> - Thank you for pointing this out, and we have fixed it in the text.
>
> Citations:
>
> [1] Chen, Ting, et al. "A simple framework for contrastive learning of visual representations." _International conference on machine learning_. PMLR, 2020.

---

### Official Review · Reviewer_mFnV · 2023-10-30

**Soundness:** 3 good
**Presentation:** 2 fair
**Contribution:** 2 fair
**Rating:** 5
**Confidence:** 3

**Summary:**

This article proposes a new perspective for determining positive and negative samples in time series contrastive learning. If one subsequence can be successfully reconstructed by retrieving information from another, it should form a positive pair. Based on this, the author trains a cross-attention module to reconstruct the masked query input subsequence from the key input subsequence. The subsequence with the lowest reconstruction error is labelled as positive, and the others are labelled as negative. Experiments show that the REBAR method of this article achieves state-of-the-art results in learning a class-discriminative embedding space.

**Strengths:**

The method proposed in this article is intuitive and easy to understand.

**Weaknesses:**

1. Format issue: All formulas in this article are not numbered. The notations in the first formula on page 5 have no corresponding definition.

2. The experimental volume of this paper is insufficient. As a new perspective in the field of time series contrastive learning, the author should validate the method on a wider dataset to demonstrate its universality for time series.

3. The article lacks discussion and analysis of key parameters. Specifically, when training the cross-attention module, what impact will the length of subsequences and the proportion of random masking have on the reconstruction effect of key subsequences? Is the reconstruction effect of the cross-attention module directly related to downstream task performance? How to set the number of candidate subsequences (proportion of positive and negative samples) when obtaining Pos/Neg labels?

**Questions:**

I noticed that when applying the REBAR metric in contrastive learning, an anchor sequence and n candidate sequences are sampled randomly. Only the candidate sequence with the smallest reconstruction loss will be determined as a positive sample of the anchor sequence. Is there a situation where, for example, when sequence A is used as an anchor sequence, the candidate sequence with the smallest reconstruction loss is sequence B, and then A and B are mutually positive samples? However, when B is used as an anchor, the candidate sequence with the smallest reconstruction loss may be another sequence C. Therefore, B and C are positive samples. But if A is also in the candidate sequences, this method will divide A into negative samples of B. This leads to conflicting conclusions when the anchor sequence is different.

---

> ### Author Response · Authors · 2023-11-21
> **Response to mFnV (1/3)**
>
> Thank you for the discussion points and suggestions. Based upon our discussion, we have significantly revised our paper to clarify the scope of our paper to be focused on time-series that can be specifically defined as a series of class-labeled subsequences, adding a new t-SNE visualization in Fig. 6 and more analysis to further expand experimental results, as well as adding a new Tbl. 3 that highlights an ablation study and hyperparameter sensitivity analysis of the REBAR model. We have addressed each of your specific points below (with W = weakness and Q = question).
>
> Formatting (W1):
>
> - Thank you. We have added numbers to all of our equations, clarified the notation, and added definitions throughout the Sec. 3 Methods and the rest of the paper. Our Sec. 3 has been completely revised and reorganized to have a better flow to improve our REBAR method's understandability.
>
> Experiment Scale (W2):
>
> - Thank you for this question. We have revised the paper to clarify the scope of our contribution: Our method targets longer (extended) time series that are themselves composed of class-labeled subsequences, such a single recording of HAR data composed of the different types of physical activities, as illustrated in the new Fig. 1. There are relatively few datasets that support this task. For example, TNC only included HAR, ECG, and a small simulated dataset. We note that our scope is well-aligned with many mobile health, embedded sensing, and IoT applications which involve continuous and passive data collection and produce very long recordings with few to no labels. This is an area where the datasets that are available for research purposes have lagged behind the driving applications. We note that datasets such as UEA and UCR contain pre-segmented class-labeled time series that are not suitable for evaluating our approach (our datasets have time-series that are 8,325,000 / 334,080 / 15,000 time points long, compared to the median length of 621 and 301 for UEA and UCR respectively). Please see revisions to Sec. 1. Introduction and Sec. 6 Conclusions.
>   - Using this type of long time-series data in our problem also allows us to benchmark sampling-based contrastive learning methods, such as TNC and our own REBAR method. These sampling-based methods exploit the time-series nature of our task. Because each subsequence is part of a longer time-series that is changing labels over time, we can sample positive and negative examples directly from the longer time-series, rather than attempting to create positive examples from augmentations.
> - With this being said, we have expanded upon our results section to increase our experimental scale to add a new Fully Supervised baseline in Tbl. 1, a new Fig. 6 t-SNE visualization, as well as a visualization of the various positive/negative pairs identified by REBAR for a given time-series in the new Fig. 7.
>   - The linear probe trained on our REBAR representation consistently achieved the strongest results, even beating the fully supervised model in PPG and HAR, achieving the same accuracies, but higher AUROC and AUPRC. This demonstrates our REBAR's methods strength in learning a representation that is better at handling class imbalance than the fully-supervised model.
>   - Our t-SNE visualizations in Fig. 6 further validate our findings regarding our REBAR method's clustering ability. In HAR, all methods except for REBAR group walk, walk up, and walk down together. In PPG, all of the methods have poor clustering, but REBAR seems to perform the best with clearer separation between classes and has fewer discontiguous regions for the same class. In ECG, REBAR continues to have the best intra-class clustering.
>   - Our positive/negative pair visualization in Fig. 7 visualizes the positive pairing from the list of candidates for a given anchor that was identified by our REBAR method, and an additional gallery of 15 more of this visualization is included in Appendix A.5. We see that even when there is no exact match of the anchor within the candidates, REBAR's motif-comparison retrieval and reconstruction is able to identify a positive example from the candidates that shares the same class as the anchor.

---

> > ### Author Response · Authors · 2023-11-21
> > **Response to mFnV (2/3)**
> >
> > Clarification on Model Design w/ Ablation Studies + Hyperparameter (W3):
> >
> > - Thank you for the discussion points: we have captured our following discussion in Appendix A.1.3. We have added a new Tbl. 3, which includes the results of our ablation study and a hyperparameter sensitivity analysis for our REBAR model to better explicate the importance of each of the model's components. Additionally, our benchmark implementation section in Appendix A.3 describes the hyperparameters associated with all of our models for reproducibility.  To address each of your questions specifically:
> > - "_Is the reconstruction effect of the cross-attention module directly related to downstream task performance?"_: Tbl. 3 shows that REBAR is robust to the hyperparameter of Number of REBAR Cross-Attn Epochs Trained. Decreasing trained REBAR cross-attention epochs from 300 to 200 increases accuracy by .006 and increases NMI by .070. Decreasing trained REBAR cross-attention epochs from 300 to 100 increases accuracy by .006 and decreases NMI by .013. Because reconstruction error decreases as epoch trained increases (as seen in Fig. 9 in the Appendix), reconstruction accuracy is not necessarily an indicator of downstream performance.
> >   - This is not a problem because the goal of our REBAR cross-attention is not to simply learn a good reconstruction, it is to learn a class-discriminative reconstruction that achieves better reconstruction when the query and key are from the same class.
> >   - We note that these results should not imply that the REBAR model should not be trained. Tbl. 5 in the Appendix shows that if REBAR is only trained for 1 epoch, then accuracy drops by 0.010 and NMI drops by 0.251.
> > - With this context, it is more informative for us to assess how each of these hyperparameters affect the quality of the trained REBAR model's reconstruction error in the context of its contrastive learning performance.
> > - "_proportion of random masking"_: The model is fairly robust to the size of the contiguous masks used for training the REBAR cross-attention. In our ablation study, compared to an initial mask size of 15, we found that using a smaller mask size of 5 time-points lowered accuracy by only 0.004 and using a larger mask size of 25 increased accuracy by .006.
> >   - For each of the datasets, the size of the contiguous mask is chosen to be large enough to mask out a potentially class-discriminative motif. This is done so that the model learns how to specifically retrieve this class-discriminative motif from the key and so that during application, class-discriminative motifs between the key and query are compared.
> >   - Thus, contiguous mask size can be tuned with a hyperparameter grid search or with expert domain knowledge. In HAR, it is a mask that covers .3 seconds (15 time-points) because the accelerometry type data typically captures information of movement from short quick jerks. In PPG, it is a mask that covers 4.69 seconds (300 time points), which we do because PPG wave morphology is relatively simple, so we would like to capture a few quasi-periods. In ECG it is a mask that covers 1.2 seconds (300 time points), which covers a heartbeat as ECG wave morphology is generally very descriptive  (e.g. QRS complex).
> >   - This also helps guide the size of the receptive field that our dilated convolutions in our cross-attention should be. In general, we tune the receptive field size to be three times the size of the contiguous mask, so that during training, the query dilated convolution would be able to effectively capture the motif surrounding the masked out query region to be compared to the key.
> >   - As we see in the table, our model is robust to the exact receptive field size. Compared to an initial receptive field of 43, a receptive field of 31 decreases accuracy by only .003 while increasing ARI by .013 and a receptive field of 55 decreases accuracy by only .016 while increasing ARI by .031.
> >
> > - "_length of subsequences"_: We would like to note that the length of subsequences is \*not\* a hyperparameter: it is tied to the experimental design. The subsequence lengths for HAR and PPG were determined by the original papers that introduced the datasets [2,3]. These works explicitly segmented their dataset into these subsequence sizes for each subsequence to be classified. In the case for ECG, the original work was not a classification task and a subsequence length was not given [4], so we used the subsequence length used in prior work on the dataset [1].
> > - "_How to set the number of candidate subsequences … ?_": The number of candidate subsequences is a hyperparameter kept constant across each of the domains, set to 20. The candidates, as well as the identified positive example for a given anchor for each class, are visualized in Fig. 7, and an additional gallery of 15 more of this visualization is included in Appendix A.5.

---

> > > ### Author Response · Authors · 2023-11-21
> > > **Response to mFnV (3/3)**
> > >
> > > "_sequence A is used as an anchor sequence … A and B are mutually positive samples … when B is used as an anchor … B and C are positive samples"_ (Q1):
> > >
> > > - This is a great point that we would like to discuss this question further in detail, with the second part of the scenario discussed in the next section. Having a "chain" of positive pairs, with A \<-\> B \<-\> C is a strength of our method that encourages intra-class clustering. The authors in [6] introduce a theoretical framework for explaining how contrastive learning methods are able to learn a class-discriminative embedding space. They argue that intra-class instances are able to be connected together through the alignment of the overlapping instances between positive pairs. Similarily in the, REBAR setting, if B is a positive example for A, and C is a positive example for B, then B would be overlapping between the two positive pairs. This would allow us to bring A and C closer together within the embedding space, potentially promoting intra-class clustering, and this is seen in our t-SNE visualization in the new Fig. 6. .
> > >
> > > "But if A is also in the candidate sequences, this method will divide A into negative samples of B" (Q1):
> > >
> > > - The idea that false negatives potentially being among the candidates (i.e. A in negative samples of B) is a great discussion point, and it is a fundamental issue within contrastive learning more broadly, rather than being specific to our REBAR method. For example, in augmentation-based approaches like SimCLR [7], they sample negative items from the batch, and the batch may contain items from the same downstream classification category as the anchor. However, they are able to still achieve strong performance. Similarly, our approach is still able to achieve the best representation learning performance across our baselines. One idea to address this could be a thresholding method, but we believe that this method would be ineffective. A threshold should not be designed to be a hyperparameter as it would be dependent on how each specific anchor's specific motifs compare to other general instances within the same class. However, learning a threshold as a function of the anchor is not obvious to do, due to the lack of class label ground truth during contrastive learning. Addressing false negatives is an ongoing research area, and we are exploring other strategies as a part of future work. We add this discussion into the Appendix A.6.
> > >
> > > Citations
> > >
> > > [1] Tonekaboni, Sana, Danny Eytan, and Anna Goldenberg. "Unsupervised representation learning for time series with temporal neighborhood coding." _arXiv preprint arXiv:2106.00750_ (2021).
> > >
> > > [2] Reyes-Ortiz, Jorge-L., et al. "Transition-aware human activity recognition using smartphones." _Neurocomputing_ 171 (2016): 754-767.
> > >
> > > [3]  Schmidt, Philip, et al. "Introducing wesad, a multimodal dataset for wearable stress and affect detection." _Proceedings of the 20th ACM international conference on multimodal interaction_. 2018.
> > >
> > > [4] Moody, George. "A new method for detecting atrial fibrillation using RR intervals." _Proc. Comput. Cardiol._ 10 (1983): 227-230.
> > >
> > > [6] Wang, Yifei, et al. "Chaos is a ladder: A new theoretical understanding of contrastive learning via augmentation overlap." _ICLR_ (2022).
> > >
> > > [7] Chen, Ting, et al. "A simple framework for contrastive learning of visual representations." _International conference on machine learning_. PMLR, 2020.

---

> > > > ### Comment · Reviewer_mFnV · 2023-11-22
> > > >
> > > > Thanks for the responses addressing the format issues and the detailed discussion on the hyperparameters. However, my concerns regarding the scale of the experiments remain, so I will keep my score as it is.

---

> ### Author Response · Authors · 2023-11-22
> **Response to expermiental scale**
>
> Thank you for your feedback to our response. We would like to emphasize that with our clarified scope on “time-series that are defined as a series of class labeled subsequences”, there are very few datasets that support this task. TNC only included HAR, ECG, as well a small simulated dataset designed specifically to assess their model's stationarity test feature [1]. Our work adds the PPG dataset for stress classification, to introduce an additional benchmark dataset for this type of time-series contrastive learning task. If you have any suggestions on additional datasets to include, we would be happy to include them in the next revision, but we believe that the three datasets we have included are very distinctive and allow us to assess the strength of each of the contrastive learning methods.
>
> HAR, PPG, and ECG each demonstrate distinctively different characteristics.  measuring acceleration, electrical activity of the heart, and blood volume change. The differences between the classes for a given dataset are all captured very differently. The activity labeled subsequences in HAR are primarily distinguished by amplitude, as seen in Fig. 10. Using PPG to identify stress is an ongoing research topic, but recent work has found that stress labeled subsequences in PPG are captured by wave shape [2]. The atrial fibrillation labeled subsequences in ECG are distinguished by irregular peak frequency. Therefore, although we only contain three datasets, their distinctive nature demonstrates the strength of our method.
>
> [1] Tonekaboni, Sana, Danny Eytan, and Anna Goldenberg. "Unsupervised representation learning for time series with temporal neighborhood coding." arXiv preprint arXiv:2106.00750 (2021).
>
> [2] Celka, Patrick, et al. "Influence of mental stress on the pulse wave features of photoplethysmograms." Healthcare technology letters 7.1 (2020): 7-12.
>
> (edited to add TNC citation)

---

### Official Review · Reviewer_jiyG · 2023-11-06

**Soundness:** 2 fair
**Presentation:** 2 fair
**Contribution:** 2 fair
**Rating:** 6
**Confidence:** 3

**Summary:**

The paper proposes a novel method for constructing positive pairs for contrastive learning in time-series data. It presents experiments across three datasets to validate the approach.

--- post rebuttal ---

I appreciate the efforts made by the authors in addressing the concerns raised in my initial review. The manuscript has undergone significant changes, resulting in notable improvements in its quality. Considering these enhancements, I have revised my score from 3 to 6.

**Strengths:**

1. The paper is motivated. The proposed method is grounded on a cogent hypothesis *"if one time-series is useful for reconstructing another, then they likely share reconstruction information and thus should be drawn together within the embedding space as positive examples."*. Essentially, the author posits that time-series with similar semantics are capable of aiding in each other's reconstruction.

2. An intriguing observation made in the paper is the difference in sparsity between cross-attention mechanisms when trained with *contiguous masks* versus *intermittent masks.*

3. The author provides a comprehensive comparison, including many relevant baselines.

**Weaknesses:**

1. The rationale for preferring a contiguous mask over an intermittent mask is presented but could be articulated with greater clarity to enhance its persuasiveness. Additionally, there seems to be some confusion regarding Figure 1. Clarification is needed as to whether the author implies that a) the contiguous mask is utilized during the training of REBAR, and b) the intermittent mask is employed when applying REBAR in contrastive learning. If this is the case, the reasons for using different masks in these contexts should be explicitly stated.

2. The experimental scale appears somewhat limited. The paper does not specify the exact number of samples within the datasets, which seem to be on the smaller side. This limitation is accentuated when compared to previous works, such as TS2VEC [1], which utilized a much larger array of datasets, including 125 from the UCR archive and 29 from the UEA archive.

3. The explanation of results requires expansion. For instance, the acronyms ARI and NMI in Table 2 are not defined within the context of the paper, leaving their significance unclear. Moreover, there is a notable difference in the results reported for the TNC on the HAR dataset between the original TNC paper [2] and this manuscript. In the origianl paper, it was reported AUPRC 0.94, Accuracy 88 while in this manuscript, it is reported AUPRC 0.98 and Accuracy 94. More information about the potential factors leading to these discrepancies would be beneficial for the reader's comprehension.


* [1] Yue, Zhihan, et al. "Ts2vec: Towards universal representation of time series." Proceedings of the AAAI Conference on Artificial Intelligence. Vol. 36. No. 8. 2022.

* [2] Tonekaboni, Sana, Danny Eytan, and Anna Goldenberg. "Unsupervised representation learning for time series with temporal neighborhood coding." arXiv preprint arXiv:2106.00750 (2021).

**Questions:**

In Table 3, the author evaluates the influence of different mask types on performance. It would be beneficial to clarify why the training stage favors a contiguous mask, while the evaluation stage shows a preference for an intermittent mask.

---

> ### Author Response · Authors · 2023-11-21
> **Response to jiyG (1/3)**
>
> Thank you for your great discussion points. Based your points, we have added 3 new figures (conv cross-attn in Fig. 3 to better explain our approach and masking procedure; t-SNE in Fig. 6 and pos pair examples in Fig. 7 to better explain our results) and a new ablation study+hyperparameter table into the main text, as well as substantially revising our paper to add more explanation of our method with a time-series motif framing, further model analysis and explanation to the results, and clarifying the scope of our paper to be focused on time-series that can be defined as series of class-labeled subsequences. We have addressed each of your specific points below (with W = weakness and Q = question).
>
> Difference in Masking (W1, Q1):
>
> - We replace the word "evaluation" with "application" here and in our text, so that it is more clear we are using the intermittent mask explicitly during application of REBAR in contrastive learning to identify positive/negative pairs. Our REBAR cross-attention model compares the motifs within a specific receptive field around a masked time-point in the query sequence with the motifs in the key sequence. Then, it retrieves the best matching motif from the key to be used for reconstruction of that specific masked time-point in the query, and this idea is illustrated in the new Fig. 1 and Fig. 3.
> - At training time, a contiguous mask is used so that the model learns to compare specific, potentially class-discriminative, motifs in the query with specific, potentially class-discriminative, motifs in the key, rather than comparing minor, transient, non-unique motifs. See Fig. 4a) and b) for attention weight visualizations.
> - At application time, a contiguous mask could be used, however, in doing so, only the motifs near the contiguously masked out region would be compared to the key. This is because when reconstructing a given masked time-point and another point contiguous to it, the receptive fields to be used to identify motifs to be compared to the key are heavily overlapping.
>   - An intermittent mask allows for many different motifs in the query, each of them captured in a receptive field around the many masked time-points dispersed throughout the signal, to be compared to the key during reconstruction. This allows for a higher coverage of the query in conducting such motif-similarity comparisons with the key. Therefore, during application, when we are testing each candidate as the key for a given anchor as the query, we would be able to identify the candidate which is most similar to the anchor as a whole.
> - We empirically justify this intuition. Table 4 in Appendix A.1.2 shows this combination of using a contiguous mask for training and an intermittent mask for application achieves the best performance in our validation experiment. Additionally, Figure 4b) demonstrates that the cross-attention model trained with a contiguous mask still maintains a sparse attention when evaluated on a transient mask. Therefore, the model is still reconstructing the query based on specific retrieved motifs from the key at each of the transiently masked out time-points.
>
> "_The paper does not specify the exact number of samples within the datasets, which seem to be on the smaller side … UCR … UEA"_ (Q2):
>
> - Thank you for the suggestion. We have added the exact number of samples within the datasets into our paper in Appendix A.2. Once we normalize our datasets by segmenting them into the subsequences that are used to train each of our self-supervised learning models, we have a total number of 76,590 / 6,914 / 1,305 sequences for ECG, HAR, and PPG, respectively. This is much larger than the datasets found in UCR and UEA: the median total dataset size in UCR is 687 and in UEA it is 621. In fact, if we look specifically at the 29 dataset subset of UEA that TS2Vec primarily presents in their paper, then this UEA subset only has a median dataset size of 412.5. Note that:
>   - 90 out of the 128 datasets in UCR and 123 out of the 183 datasets in UEA have 1,000 or less sequences.
>   - 8 out of the 128 total datasets in UCR and 14 out of the 183 total datasets in UEA have 100 or less sequences.

---

> > ### Author Response · Authors · 2023-11-21
> > **Response to jiyG (2/3)**
> >
> > Experimental Scale (Q2):
> >
> > - Thank you for this question. We have revised the paper to clarify the scope of our contribution: Our method targets longer (extended) time series that are themselves composed of class-labeled subsequences, such a single recording of HAR data composed of the different types of physical activities, as illustrated in the new Fig. 1. There are relatively few datasets that support this task. For example, TNC only included HAR, ECG, and a small simulated dataset. We note that our scope is well-aligned with many mobile health, embedded sensing, and IoT applications which involve continuous and passive data collection and produce very long recordings with few to no labels. This is an area where the datasets that are available for research purposes have lagged behind the driving applications. We note that datasets such as UEA and UCR contain pre-segmented class-labeled time series that are not suitable for evaluating our approach (our datasets have time-series that are 8,325,000 / 334,080 / 15,000 time points long, compared to the median length of 621 and 301 for UEA and UCR respectively). Please see revisions to Sec. 1. Introduction and Sec. 6 Conclusions.
> > - Using this type of long time-series data in our problem also allows us to benchmark sampling-based contrastive learning methods, such as TNC and our own REBAR method. These sampling-based methods exploit the time-series nature of our task. Because each subsequence is part of a longer time-series that is changing labels over time, we can sample positive and negative examples directly from the longer time-series, rather than attempting to create positive examples from augmentations.
> >
> > Detailed Explanation of Results (Q3):
> >
> > - Thank you for this point, and we have significantly revised our Sec. 5 Results section to include a new fully-supervised baseline in Tbl. 1, a new Fig. 6 t-SNE visualization and a REBAR approach analysis subsection.
> > - The linear probe trained on our REBAR representation consistently achieved the strongest results, even beating the fully supervised model in PPG and HAR, achieving the same accuracies, but higher AUROC and AUPRC. This demonstrates our REBAR's methods strength in learning a representation that is better at handling class imbalance than the fully-supervised model.
> > - Our t-SNE visualizations in Fig. 6 further validate our findings regarding our REBAR method's clustering ability. In HAR, all methods, except for REBAR, group "walk", "walk up", and "walk down" together. In PPG, all of the methods have poor clustering, but REBAR seems to perform the best with clearer separation between classes and has fewer discontiguous regions for the same class. In ECG, REBAR continues to have the best intra-class clustering.
> > - The REBAR approach analysis contains a new Tbl. 3 with the results of our ablation study and a hyperparameter sensitivity analysis, as well as a visualization of the various positive/negative pairs identified by REBAR for a given time-series in the new Fig. 7.
> >   - Each of the model components and modifications contributes to REBAR's strong contrastive learning performance, especially the dilated convolutions (e.g. .061 drop in accuracy and a .263 drop in NMI). The exclusion of our reversible instance norm leads to a .008 drop in accuracy and a .127 drop in NMI, and the addition of an explicit positional embedding leads to a .006 drop in accuracy and .120 drop in NMI. As noted in Section 3.1, we intentionally keep the design of REBAR attention simple to emphasize the motif comparison within the cross-attention mechanism.
> >   - The REBAR model is fairly robust to hyperparameter tuning. When we modify the size of contiguous masks,  the size of the dilated conv's receptive field, or the number of epochs to train the REBAR cross-attention, the downstream linear probe and clusterability performance remains consistent.
> >     - Increasing the mask size from 15 to 25 increases accuracy by .006 and increases NMI by .017. Decreasing the mask size from 15 to 5 decreases accuracy by .004 and decreases NMI by .017.
> >     - Increasing the receptive field from 43 to 55 decreases accuracy by .016 and increases NMI by .037. Decreasing the receptive field from 43 to 31 decreases accuracy by .003 and increases NMI by .007
> >     - Decreasing trained REBAR cross-attention epochs from 300 to 200 increases accuracy by .006 and increases NMI by .070. Decreasing trained REBAR cross-attention epochs from 300 to 100 increases accuracy by .006 and decreases NMI by .013.
> >     - We note that these results should not imply that the REBAR model should not be trained. Tbl. 5 in the Appendix shows that if REBAR is only trained for 1 epoch, then accuracy drops by 0.010 and NMI drops by 0.251.

---

> > > ### Author Response · Authors · 2023-11-21
> > > **Response to jiyG (3/3)**
> > >
> > > [CONTINUED] Detailed Explanation of Results (Q3):
> > >   - In Fig. 7 positive/negative pair visualization, we illustrate the positive example identified by our REBAR method from the list of candidates, for a given anchor in each class. An additional gallery of 15 more of this visualization is included in Appendix A.5. Even when there is no exact match for the anchor within the candidates, REBAR is still able to select an example that has the same class as the anchor.
> > >
> > > "_ARI and NMI in Table 2 are not defined"_ (Q3):
> > >
> > > - Thank you; we have replaced the acronyms with their full names, Adjusted Rand Index and Normalized Mutual Information, in Sec. 4 Experimental Design and in the caption of Tbl. 2 clusterability results. These two metrics are commonly used to evaluate a representation's clusterability by assessing how similar k-mean clusters of the encoding are to the true class labels [1-3].
> > >
> > > "_difference in the results reported for the TNC on the HAR"_ (Q3):
> > >
> > > - Our reported TNC results for HAR are higher than the original reported results because, we use TS2Vec's proposed dilated convolution encoder, and in order to fairly compare all of the contrastive learning baselines, we keep this encoder backbone constant across all of the baselines. In TNC's original work, they used a simpler RNN model for the HAR data. TSVec also found that training TNC with the dilated convolution encoder achieves better results (See Appendix C.3 in [4]). Additionally, other discrepancies may originate from how TNC evaluated their method by training a classification head with the encoder model end-to-end, rather than freezing the encoding. We opt to freeze the encoder in order to allow for the encoded representation to be directly evaluated via a linear classifier (i.e. Acc, AUROC, AUPRC) and clusterability (i.e. Adjusted Rand Index and Normalized Mutual Information). We have added these details into the Appendix A.3
> > >
> > > Citations
> > >
> > > [1] Hassani, Kaveh, and Amir Hosein Khasahmadi. "Contrastive multi-view representation learning on graphs." _International conference on machine learning_. PMLR, 2020.
> > >
> > > [2] Fan, Xiaolong, et al. "Maximizing mutual information across feature and topology views for learning graph representations." arXiv preprint arXiv:2105.06715 (2021).
> > >
> > > [3] Zhang, Xiang, et al. "Self-supervised contrastive pre-training for time series via time-frequency consistency." Advances in Neural Information Processing Systems 35 (2022): 3988-4003.
> > >
> > > [4] Yue, Zhihan, et al. "Ts2vec: Towards universal representation of time series." _Proceedings of the AAAI Conference on Artificial Intelligence_. Vol. 36. No. 8. 2022. [arxiv.org/pdf/2106.10466.pdf](https://arxiv.org/pdf/2106.10466.pdf)

---

> > > > ### Comment · Reviewer_jiyG · 2023-11-22
> > > > **Response to Response (3/3)**
> > > >
> > > > Thanks for the explaination. My issue on acronyms and difference in the results are addressed.

---

> > ### Comment · Reviewer_jiyG · 2023-11-22
> > **Response to Response (1/3)**
> >
> > I appreciate the author's effort on addressing my issue on "Difference in Masking". I am satisfied with the modification the author have made.

---

### Official Review · Reviewer_gCTH · 2023-11-06

**Soundness:** 2 fair
**Presentation:** 3 good
**Contribution:** 2 fair
**Rating:** 5
**Confidence:** 4

**Summary:**

This paper introduces a novel approach called Retrieval-Based Reconstruction (REBAR) for self-supervised contrastive learning in time-series data. The REBAR method utilizes retrieval-based reconstruction to identify positive data pairs in time-series, leading to state-of-the-art performance on downstream tasks.

**Strengths:**

1. Novel approach: The paper introduces a novel approach called Retrieval-Based Reconstruction (REBAR) for self-supervised contrastive learning in time-series data. This approach utilizes retrieval-based reconstruction to identify positive data pairs in time-series, which is a unique and effective way to address the challenges of creating positive pairs via augmentations in time-series data.

2. State-of-the-art performance: The paper demonstrates that the REBAR method achieves state-of-the-art performance on downstream tasks across diverse modalities, including speech, motion, and physiological data.

3. Comprehensive evaluation on two tasks including classification and cluster agreement.

**Weaknesses:**

1. Lack of ablation studies: The paper does not include ablation studies to analyze the contribution of each component of the REBAR method. This makes it difficult to understand the relative importance of each component and how they interact with each other.

2. Detailed explanation of the results. There is no detailed studies for table 1 and 2, e.g., visualizations of the learned embedding or positive/negative pairs.

3. Limited discussion of hyperparameters: While the paper provides some details about the hyperparameters used in the experiments, it does not provide a comprehensive analysis of the sensitivity of the method to different hyperparameters.

4. Without comparison with baselines, Figure 4 doesn't show any advantages of the proposed model since the diagonal pattern would be obvious for most of the baselines.

5. Section 3.1, notations are used without clear definition

**Questions:**

1. Ablation study and hyperparameters selection.
2. Include more visualizations or examples of the positive and negative pairs identified by the REBAR method
3. "During evaluation, we use an an intermittent mask", explain the intuition why different masks are used in the training and evaluation
4. How does the REBAR method perform on time-series data with different characteristics, such as varying lengths or noise levels?
5. Provide more detailed explanations of the convolutional cross-attention architecture used in the REBAR method.

---

> ### Author Response · Authors · 2023-11-21
> **Response to gCTH (1/2)**
>
> Thank you for your great suggestions. We have added 5 new figures (intuition in Fig. 1; conv cross-attn in Fig. 3; t-SNE in Fig. 6; pos pair in Fig. 7) and a new ablation study+hyperparameter table into the main text, as well as substantially revising our paper to capture your comments. We have addressed each of your points below (with W = weakness and Q = question).
>
> Lack of Ablation Studies + Hyperparameter Analysis (W1, W3, Q1):
>
> - Thank you for this suggestion: we have added a new Table 3, which includes the results of our ablation study and a hyperparameter sensitivity analysis for our REBAR model.
> - Each of the model components and modifications contributes to REBAR's strong contrastive learning performance, especially the dilated convolutions (e.g. .061 drop in accuracy and a .263 drop in NMI). The exclusion of our reversible instance norm leads to a .008 drop in accuracy and a .127 drop in NMI, and the addition of an explicit positional embedding leads to a .006 drop in accuracy and .120 drop in NMI. As noted in Section 3.1, we intentionally keep the design of REBAR attention simple to emphasize the motif comparison within the cross-attention mechanism.
> - The REBAR model is fairly robust to hyperparameter tuning. When we modify the size of contiguous masks,  the size of the dilated conv's receptive field, or the number of epochs to train the REBAR cross-attention, the downstream linear probe and clusterability performance remains consistent.
>   - Increasing the mask size from 15 to 25 increases accuracy by .006 and increases NMI by .017. Decreasing the mask size from 15 to 5 decreases accuracy by .004 and decreases NMI by .017.
>   - Increasing the receptive field from 43 to 55 decreases accuracy by .016 and increases NMI by .037. Decreasing the receptive field from 43 to 31 decreases accuracy by .003 and increases NMI by .007
>   - Decreasing trained REBAR cross-attention epochs from 300 to 200 increases accuracy by .006 and increases NMI by .070. Decreasing trained REBAR cross-attention epochs from 300 to 100 increases accuracy by .006 and decreases NMI by .013.
>   - We note that these results should not imply that the REBAR model should not be trained. Tbl. 5 in the Appendix shows that if REBAR is only trained for 1 epoch, then accuracy drops by 0.010 and NMI drops by 0.251.
>
> More Visualizations of Learned Embedding + Positive/Negative pairs. (W2, Q2)
>
> - Thank you for this idea. In the Sec. 5 Results section, we have added t-SNE visualizations of each model's encoding in the new Fig. 6, and a visualization of the various positive/negative pairs identified by REBAR for a given time-series in new Fig. 7, as well as an additional gallery of 15 more of this positive/negative pair visualization within Appendix A.5.
> - Our t-SNE visualizations in Fig. 6 further validate our findings regarding REBAR's ability to learn embeddings with superior class-discriminative properties. In HAR, all methods, except for REBAR, group "walk", "walk up", and "walk down" together. In PPG, all of the methods have poor clustering, but REBAR seems to perform the best with clearer separation between classes and fewer non-contiguous regions within the same class. In ECG, REBAR exhibits the best intra-class clustering.
> - In Fig. 7 positive/negative pair visualization, we illustrate the positive example identified by our REBAR method from the list of candidates, for a given anchor in each class. Even when there is no exact match for the anchor within the candidates, REBAR is still able to select an example that has the same class as the anchor.
>
> "_Without comparison with baselines, Figure 4 [confusion matrices] …"_ (W4) :
>
> - We would like to clarify that the confusion matrices (now in Fig. 5) are designed to assess how REBAR's pseudo-distance function between two subsequences predicts mutual class membership between the two. This validates REBAR's effectiveness in learning a discriminative distance function, before contrastive learning takes place. Since the other baselines from the literature do not construct a pseudo-distance function, they can't be included in this analysis. We have clarified the text to make this more clear.
>
> Notation (W5):
>
> - Thank you: we have added notation definitions at the start of the Sec. 3 Methods section and clarified the notation further in 3.1 with more details.

---

> ### Author Response · Authors · 2023-11-21
> **Response to gCTH (2/2)**
>
> "_intuition why different masks are used in the training and evaluation [i.e. application]"_ (Q3)
>
> - We replace the word "evaluation" with "application" here and in our text, so that it is more clear we are using the intermittent mask explicitly during application of REBAR in contrastive learning to identify positive/negative pairs. Our REBAR cross-attention model compares the motifs within a specific receptive field around a masked time-point in the query sequence with the motifs in the key sequence. Then, it retrieves the best matching motif from the key to be used for reconstruction of that specific masked time-point in the query, and this idea is illustrated in the new Fig. 1 and Fig. 3.
> - At training time, a contiguous mask is used so that the model learns to compare specific, potentially class-discriminative, motifs in the query with specific, potentially class-discriminative, motifs in the key, rather than comparing minor, transient, non-unique motifs. See Fig. 4a) and b) for attention weight visualizations.
> - At application time, a contiguous mask could be used, however, in doing so, only the motifs near the contiguously masked out region would be compared to the key. This is because when reconstructing a given masked time-point and another point contiguous to it, the receptive fields to be used to identify motifs to be compared to the key are heavily overlapping.
>   - An intermittent mask allows for many different motifs in the query, each of them captured in a receptive field around the many masked time-points dispersed throughout the signal, to be compared to the key during reconstruction. This allows for a higher coverage of the query in conducting such motif-similarity comparisons with the key. Therefore, during application, when we are testing each candidate as the key for a given anchor as the query, we would be able to identify the candidate which is most similar to the anchor as a whole.
> - We empirically justify this intuition. Table 4 in Appendix A.1.2 shows this combination of using a contiguous mask for training and an intermittent mask for application achieves the best performance in our validation experiment. Additionally, Figure 4b) demonstrates that the cross-attention model trained with a contiguous mask still maintains a sparse attention when evaluated on a transient mask. Therefore, the model is still reconstructing the query based on specific retrieved motifs from the key at each of the transiently masked out time-points.
>
> REBAR performance on Distinctive Time-series Domains (Q4)
>
> - We test our approach in three different time series domains which consist of signals with qualitatively very different properties, as illustrated in the new Fig. 10-12. The different classes within HAR are arguably more distinctive than the classes for ECG and PPG, although the different HAR categories of "walk", "walk up", and "walk down" require a more subtle distinction. This is reflected in our results, with our method performing the best on HAR. The fact that we achieve SOTA results in three very different data domains speaks to the effectiveness of our method. Regarding more systematic variations, we note that our datasets also contain substantial variation in signal lengths. The subsequence lengths are adapted to each of the signal domains and vary significantly in length, and the source signals that they are sampled from vary significantly as well. HAR uses 2.56s subsequences sampled from a 5 min time-series. PPG uses 60s subsequences sampled from an 87 min time-series. ECG uses 10s subsequences sampled from a 9.25 hour time-series.
>
> More explanation of REBAR Convolutional Cross-Attention (Q5):
>
> - Thank you for this comment. In Sec. 1 Introduction, we have added the new time-series motif language to help better explain the intuition of our idea, and a new Fig. 1 has been added to illustrate the intuition. Sec. 3 Methods has been completely revised and re-organized to incorporate the motif language. The start of Sec. 3 has been modified to introduce the idea of using REBAR's reconstruction error as a distance metric earlier, motivating our approach. Sec. 3.1 now ties the mathematical formulation of cross-attention with our new Fig. 3 to explicitly demonstrate how convolutions aid the cross-attention in retrieving and comparing motifs. Training REBAR is now its own section, Sec. 3.2, with Applying REBAR is Sec. 3.3. In these two sections, we are able to ground the masking motivation more clearly with the motif language. With the ablation study and hyperparameter sensitivity results in Tbl. 3, we can clearly understand how each of the REBAR cross-attention components contribute to downstream performance, as well as how to train the model. Finally, in the Appendix A.1, we include extra details on modeling decisions, as well as a new Fig. 8 that illustrates the exact dilated convolution module used in REBAR.

---

### Author Response · Authors · 2023-11-21
**Response to all Reviewers**

We would like to thank all of our reviewers for their insightful comments to help improve the paper. In response to the feedback and suggestions, we:

- Added new experiments, including demonstrating that a simple linear classifier with our REBAR model's frozen encoding can beat a fully-supervised baseline in PPG and HAR, a new t-SNE Fig. 6 that visualizes REBAR's clusterability strengths, and a new Tbl. 3 that covers an ablation study with hyperparameter sensitivity analysis.
- Added 3 new figures (Fig. 1 to illustrate the idea of our approach, Fig. 3 to demonstrate how the dilated convolutions help in retrieval and reconstruction within cross-attention, and Fig. 7 to show examples of positive pairs identified by REBAR) and revised our paper to include a time-series motif framing to better explain how our REBAR cross-attention model and masking approaches work.
- Improved the writing and clarified our focus on time-series that are specifically composed of a series of class-labeled subsequences, rather than general time-series.  Our focus aligns with the time series data produced by modern wearable and embedded sensing applications (such as continuous data collection from body-worn physiological sensors).

Please see the specific responses to each individual reviewer below. We would be happy to address further questions.

---

### Meta-Review · Area_Chair_nEbs · 2023-12-12

**Metareview:**

This paper proposes a retrieval-based method for contrastive learning with time series data. The method is used to learn embeddings. The authors show that these embeddings are useful for achieving performance on downstream tasks that the reviewers agree are state of the art on downstream tasks. The paper received mixed reviews, with two recommending acceptance and two rating the paper as a borderline reject. Among the dissenters, one reviewer declined the opportunity to reply to the author's rebuttal while the other replied but stuck by their score mostly on account of concerns over the scale of experiments. This concern about the scale of the experiments was shared among other reviewers, including jiyG who rated the paper as an Accept. The strongest score is an 8, although this review comes in underweight and anemic.

**Justification For Why Not Higher Score:**

Limited experimental evaluation.

**Justification For Why Not Lower Score:**

Interesting methodological contribution, nice experimental results (even if small in scale), effort to provide missing ablations in the rebuttal.

---

### Decision · Program_Chairs · 2024-01-16

Accept (poster)